# A Practitioner's Guide to Multi-turn Agentic Reinforcement Learning

## Abstract

We study what actually works and what doesn't for training large language models as agents via multi-turn reinforcement learning. Despite rapid progress, existing frameworks and definitions are fragmented, and there is no systematic formulation or analysis of which design choices matter across tasks. We address this gap by first breaking down the design space into three inter-related pillars—environment, reward, and policy—and empirically derive a recipe for training LLM agents in situated textual domains. In particular, we test TextWorld and ALFWorld, popular domains for testing situated embodied reasoning, as well as SWE-Gym for more software engineering style tasks. (i) For the environment, we analyze the impacts of task complexity in terms of sizes of the state and action spaces as well as optimal solution length, finding that even simple environments within a domain can provide signal on how well an agent can generalize to more complex tasks. (ii) For the reward, we ablate relative reward sparsity, observing that while dense turn-level rewards accelerate training, performance and stability is highly dependent on the choice of RL algorithm. (iii) And for the agent's policy, we explore the interplay between reward sparsity and biased (PPO, GRPO) and unbiased (RLOO) policy gradient methods in addition to showing how to find the optimal Supervised Fine-tuning (SFT) to RL training ratio given a fixed budget.

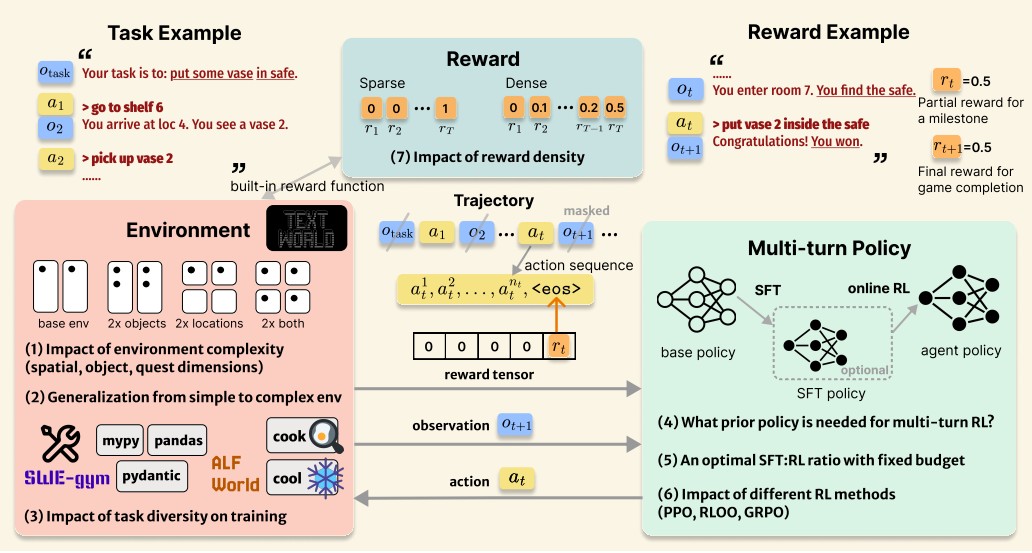

Figure 1: Illustration of multi-turn agentic RL and the key research questions.

# 1 INTRODUCTION

Training LLMs as autonomous agents to navigate open-ended environments presents unique challenges: planning across extended horizons, making multi-turn sequential decisions, and optimizing for multi-turn rewards. The transition from static single-turn problem-solving to dynamic multi-step reasoning is essential for agentic benchmarks such as interactive text and embodied simulations (TextWorld (Côté et al., 2018), ALFWorld (Shridhar et al., 2021), etc.), real-world software programming (OSWorld (Xie et al., 2024), SWE-gym (Pan et al., 2025), etc.), and abstract reasoning in novel situations (ARC-AGI (Chollet et al., 2025)). However, existing multi-turn RL implementations vary widely: some refer to tool-augmented single queries as multi-turn (Zeng et al., 2025), while many rely on model-based assumptions (Wang et al., 2025). This fragmentation has led to incomparable results across papers and confusion about what constitutes true multi-turn learning versus pseudo-multi-turn adaptations of single-turn methods.

This paper aims to facilitate research efforts on the open research question: *What factors are practically important in making multi-turn RL for LLM agent learning work*. Motivated by the lack of standardization of multi-turn RL approaches, we systematically decompose the design space into three interdependent pillars—environment, reward, and policy—and empirically derive a recipe for training LLM agents in situated textual domains (Figure 1). We evaluate our approach on TextWorld and ALFWorld for embodied reasoning, and SWE-gym for real-world programming, revealing critical insights for each pillar. For the **environment**, we investigate how performance scales with environment complexity and task diversity, and how agents generalize across different environments and tasks. For the **policy**, we investigate how model priors affect continual multi-turn RL training and analyze the interplay between multi-turn imitation learning (SFT) and multi-turn RL. We further compare biased (PPO, GRPO) and unbiased (RLOO) policy gradient RL algorithms to isolate benefits from algorithmic heuristics. For the **reward**, we experiment with varying densities of per-turn rewards to understand their impact on training.

The experiments show that our recipe works across textual reasoning, situated embodied reasoning, as well as software engineering tasks. The key findings from our analysis are: (1) Multi-turn RL performance scales with environment complexity across spatial, object, and solution dimensions; (2) Agents trained on simpler environments show promising generalization to complex ones; (3) Multi-task training enhances multi-turn RL performance; (4) Model priors from minimal demonstrations accelerate convergence, but RL remains essential for generalization; (5) An optimal SFT:RL ratio exists that balances task accuracy and generalization under fixed budgets; (6) Both biased and unbiased algorithms achieve stable learning, validating that gains stem from our multi-turn formulation rather than algorithmic heuristics; (7) Dense turn-level rewards accelerate multi-turn RL training compared to sparse rewards, but require algorithm-specific tuning.

These insights yield a concrete multi-turn RL recipe that guides co-design across all three pillars. We demonstrate that multi-turn RL with LLMs is not simply an extension of single-turn optimization but requires fundamental rethinking across environment, policy, and reward. To facilitate future research, we will release our multi-turn agentic RL framework built on veRL (Sheng et al., 2025), including the training scripts and model weights of the experiments in this paper. This work provides both theoretical insights and practical guidelines for developing agentic AI systems capable of operating effectively in real-world interactive environments.

# 2 RELATED WORK

While single-turn RL methods for LLMs including PPO (Schulman et al., 2017), RLOO (Ahmadian et al., 2024), GRPO (Shao et al., 2024), and DAPO (Yu et al., 2025) have been extensively optimized for immediate response quality, adapting them to multi-turn agentic scenarios remains non-trivial. These methods assume rewards directly follow individual actions, but multi-turn environments only reveal outcomes after extended interaction sequences, breaking the action-reward coupling that single-turn methods rely upon. Existing efforts on multi-turn RL have made limited progress on these challenges. Some approaches construct multi-turn scenarios by interleaving tool-use or reasoning steps for single-turn QA pairs (Zeng et al., 2025; Dong et al., 2025). Others who work on true interactive environments either rely on sparse terminal rewards without turn-level learning signals (Wang et al., 2025), or assign turn-level advantages uniformly across sequence tokens without fine-grained credit assignment (Zhou et al., 2025). More importantly, there lacks a

comprehensive understanding of how the three fundamental pillars of RL – environment, policy, and reward – jointly determine performance in multi-turn interactive environments. This paper provides a systematic analysis on how the fundamental pillars of RL impact multi-turn RL training, respectively, and concludes insights on how to practically train multi-turn RL in different interactive agentic environments. Throughout the paper, we dedicate related works in individual sections.

## 3 MULTI-TURN AGENTIC REINFORCEMENT LEARNING

We formulate multi-turn agentic tasks as a Partially Observable Markov Decision Process (POMDP) problem, defined as a tuple $(\mathcal{S}, \mathcal{A}, \mathcal{T}, \mathcal{R}, \Omega, \mathcal{O}, \gamma)$. Taking the Textworld task (Côté et al., 2018) as an example, an agent takes the action $a_t$ (go south) sampled from the action space $\mathcal{A}$ and receives a text observation $o_t$ (You are in front of a garden) from the observation space $\Omega$. $o_t$ is a partial description of the true state $s_t$ in the hidden state space $\mathcal{S}$ which contains the complete state world model. We assume that the state transition function $\mathcal{T} : \mathcal{S} \times \mathcal{A} \rightarrow \mathcal{S}$ is deterministic. Upon taking an action, the agent also receives a scalar reward $r_t = \mathcal{R}(s_t, a_t)$. The agent's objective is to learn a policy that maximizes the expected discounted sum of rewards $\mathbb{E}[\sum_t \gamma^t \cdot r_t]$.

We denote the trajectory history consisting of a task prompt $u$, action and state sequences by $h_t = (u, s_0, a_0, s_1, a_1, \cdots, s_t)$[1]. An LLM agent with policy $\pi_\theta$ samples an action sequence $a_t \sim \pi_\theta(\cdot|h_t)$ based on the trajectory history. $a_t$ is a token sequence in natural language: $(a_t^1, a_t^2, ..., a_t^{n_t}, a_t^{eos})$, with each token $a_t^i$ generated as $\pi_\theta(\cdot|h_t, a_t^{<i})$. Agentic environments execute language commands only upon completion, naturally defining the reward structure at the command boundaries, marked by <eos> tokens. Therefore, we assign scalar reward $r_t$ at $a_t^{eos}$, and the reward for each action token is formulated as: $r_t^i = \begin{cases} r_t & \text{if } a_t^i = \text{<eos>} \\ 0 & \text{otherwise} \end{cases}$. We make sure only action tokens contribute to the loss by masking out all state tokens.

Here is a concrete example: the input to the LLM during the rollout stage using a chat template is:

```
<|im_start|>user
Your task is: {task prompt}. state: {state 0} your action:<|im_end|>
<|im_start|>assistant
{action 0}<|im_end|>
...
<|im_start|>user
state: {state t} your action:<|im_end|>
<|im_start|>assistant
```

The LLM of policy $\pi_\theta$ generates the output {action t}<|im_end|>. The environment handles state transition and reward computation: next_state, reward, done = env.step(state, action). The reward for each turn is assigned to the <|im_end|> token. The action and next state are then appended to the chat history under the template.

## 4 BACKGROUND AND EXPERIMENTAL SETUP

Our experiments systematically investigate how three fundamental pillars of RL impact multi-turn agentic RL performance. For **Environment** (§5), we examine how scaling environment complexity affects learning across spatial, object, and solution dimensions (§5.1), evaluate whether agents trained on simpler environments generalize to complex ones (§5.2), and analyze how task diversity impacts training and generalization (§5.3). For **Policy** (§6), we analyze how the model priors from demonstration data influence RL convergence and identify optimal ratios of SFT-to-RL data ratios under budget constraints (§6.1). We compare biased algorithms (PPO, GRPO) against unbiased algorithms (RLOO) to isolate benefits of algorithmic design versus our multi-turn formulation (§6.2). For **Reward** (§7), we investigate how reward density, the frequency of feedback signals during trajectories, affects training and final performance (§7.1). These experiments demonstrate that multi-turn RL requires careful co-design across all components rather than naive extensions of single-turn methods.

---

[1]We substitute observation $o$ for state $s$ for simplicity. The agent has no access to the true state of the game.

**Tasks and Environments.**   We evaluate on three situated textual benchmarks: TextWorld (Côté et al., 2018) and ALFWorld (Shridhar et al., 2021) for language grounding in situated embodied reasoning, and SWE-Gym (Pan et al., 2025) for real-world software engineering. We extend veRL (Sheng et al., 2025)'s efficient RL training infrastructure, integrating these benchmark environments through standard `step` and `reset` interfaces. Critically, unlike traditional RL settings that provide both observations and admissible action lists (reducing the problem to action selection), our agents must generate executable natural language commands from environment observations alone, without action hints—testing their ability to discover valid actions through exploration. The details of tasks and environments are provided in Appendix B.

- **TextWorld**: A text adventure game environment where agents navigate rooms, manipulate objects, and solve quests via natural language. We procedurally generate tasks with controlled complexity across three dimensions: world size (w), number of objects (o), and quest length (q). For example, "w2-o3-q4" denotes 2 rooms, 3 objects, and a 4-step quest. Each task is uses unique seeds for diversity.

- **ALFWorld**: An embodied household environment requiring multi-step task completion through text interaction. We use the text-only variant spanning six task categories, training on the "train" split and evaluating on the "valid_unseen" split.

- **SWE-Gym**: Real-world programming tasks including bug fixes and feature implementation. We focus on 5 representative task types: getmoto, pydantic, mypy, pandas, and iterative_dvc, randomly sampling 90 training and 25 evaluation instances.

**Training and Evaluation.**   We use Qwen2.5-1.5B-Instruct, Qwen2.5-7B-Instruct, and Qwen3-8B as base models (abbreviated as Qwen-1.5B, Qwen-7B, and Qwen-8B), training with PPO (Schulman et al., 2017), GRPO (Shao et al., 2024), and RLOO (Ahmadian et al., 2024) algorithms. Maximum iteration steps and token limits scale with task complexity. During rollout generation, we use temperature 0.7 to balance exploration-exploitation. We evaluate agents on held-out test sets, reporting task success rate as our primary metric—the percentage of episodes where agents achieve objectives within exploration budgets. For SWE-Gym, we report test suite passing ratios. Full training details and hyperparameter tuning analysis appear in Appendices C and A.

## 5 ENVIRONMENT

The environment fundamentally determines the challenges agents must overcome. While single-turn tasks primarily measure reasoning difficulty, multi-turn environments introduce dimensions such as spatial navigation, object manipulation, and extended planning horizons. We investigate three core research questions for practical multi-turn deployment: (1) *How does environment complexity affect multi-turn RL training efficiency?* This determines exploration budgets and model size requirements for tasks with varied environment complexities. (2) *Can agents trained on simpler environments generalize to complex ones?* This addresses the key consideration to scaling up agentic systems. (3) *How does task diversity impact training and generalization?* We not only show our multi-turn RL recipe that works for TextWorld generalizes to ALFWorld and SWE-Gym, but also reveal whether agents learn generalizable skills or memorize task-specific behaviors. Our experiments demonstrate that multi-turn RL enables effective knowledge transfer across environments and diverse tasks.

**Setup.**   We design two experimental conditions to vary environment complexity and task diversity. For **environment complexity**, starting from the base configuration w2-o3-q4, we create controlled variations: w8-o3-q4 (increased spatial complexity), w2-o12-q4 (increased object complexity), w8-o12-q4 (increased both dimensions), and w4-o6-q8 (proportional scaling across all dimensions). In SWE-Gym, we focus on the getmoto task subset to isolate complexity factors, categorizing tasks into Easy, Medium, and Hard based on two distinct metrics: (1) *Solution Complexity* (lines of code in the gold patch) and (2) *Verification Complexity* (size of the test suite). For **task diversity**, we construct mixtures of increasing heterogeneity for training. In ALFWorld, we test single type (pick & place only), 4 types (heat, cool, cook, examine) and all 6 types (adding pick two & place). In SWE-Gym, we compare single type (getmoto only) against all 5 types (getmoto, pydantic, mypy, pandas, and iterative_dvc). All mixtures have the same data size to ensure fair comparison, with evaluation on both single-type and mixed-types held-out test sets. We train Qwen-1.5B, Qwen-7B, and Qwen-8B models using PPO and GRPO with consistent hyperparameters. Full task specifications and training details appear in Appendices B.1 and C.1.

## 5.1 HOW DOES MULTI-TURN RL PERFORMANCE SCALE WITH ENVIRONMENT COMPLEXITY?

Table 1 reveals that base models struggle dramatically as environment complexity increases, with performance dropping from 17% to just 3% when both spatial and object dimensions are scaled. Critically, **multi-turn RL improvements diminish with increasing complexity**—while PPO achieves a 88% improvement on the base environment, this drops to 51% on the most complex setting. Notably, **object complexity proves more challenging than spatial complexity**, suggesting that manipulating and tracking multiple objects across turns presents fundamentally harder challenges than spatial exploration in situated textual domains. We observe parallel trends in the coding domain (Table 2). Base model performance degrades as tasks require longer code solutions or pass more rigorous test suites. However, Multi-turn GRPO consistently recovers performance across all difficulty tiers, validating that our metrics capture genuine complexity.

**Performance also scales with model size**. As shown in Table 3, the 7B model reaches 72% success on w4-o6-q8, demonstrating that larger models better handle the increased state spaces of complex environments. Notably, even the 1.5B model shows strong learning potential with 65% and 58% gains on w2-o3-q4 and w4-o6-q8 respectively.

Table 1: Multi-turn PPO performance on TextWorld tasks with varying complexity dimensions. Base environment: w2-o3-q4. Maximum steps: 12.

| Tasks w/ varying env complexity | Qwen-1.5B | Qwen-1.5B (PPO) |
|---|---|---|
| w2-o3-q4 (base env) | 0.17 | $0.88_{\uparrow 0.71}$ |
| w8-o3-q4 (4x rooms) | 0.07 | $0.68_{\uparrow 0.61}$ |
| w2-o12-q4 (4x objects) | 0.08 | $0.54_{\uparrow 0.46}$ |
| w8-o12-q4 (4x objects & rooms) | 0.03 | $0.51_{\uparrow 0.48}$ |

Table 2: Multi-turn GRPO performance on SWE-Gym tasks across varying complexity dimensions. We compare the Qwen3-8B base model against the policy trained with Multi-turn GRPO, categorizing tasks by solution size (lines of patch) and verification effort (test suite size).

| Task Complexity Metrics | Qwen3-8B (Base) | Qwen3-8B (GRPO) |
|---|---|---|
| *Metric A: Solution Complexity (Lines of Code)* | | |
| Easy ($\leq$ 8 lines) | 0.048 | $0.072_{\uparrow 0.024}$ |
| Medium (9–31 lines) | 0.048 | $0.109_{\uparrow 0.061}$ |
| Hard ($>$ 31 lines) | 0.018 | $0.096_{\uparrow 0.078}$ |
| *Metric B: Verification Complexity (Test Suite Size)* | | |
| Easy ($\leq$ 17 tests) | 0.040 | $0.133_{\uparrow 0.093}$ |
| Medium (18–53 tests) | 0.028 | $0.075_{\uparrow 0.047}$ |
| Hard ($>$ 53 tests) | 0.017 | $0.105_{\uparrow 0.088}$ |

Table 3: Multi-turn PPO performance comparison across model sizes on proportionally scaled TextWorld environments. Maximum steps: 12 (w2-o3-q4) and 24 (w4-o6-q8).

| Tasks | Qwen-1.5B | Qwen-1.5B (PPO) | Qwen-7B | Qwen-7B (PPO) |
|---|---|---|---|---|
| w2-o3-q4 | 0.15 | $0.8_{\uparrow 0.65}$ | 0.65 | $0.98_{\uparrow 0.33}$ |
| w4-o6-q8 | 0.01 | $0.59_{\uparrow 0.58}$ | 0.28 | $0.72_{\uparrow 0.44}$ |

Lastly, we investigate how the exploration budget (the maximum steps agents can take during rollout) affects learning. For w2-o3-q4 tasks with 4-step optimal solutions, we vary maximum steps from 6 to 16. Table 4 shows that **performance saturates beyond 8 exploration steps**. Constraining agents to 6 steps ($1.5\times$ optimal) limits PPO to 55% success, while 8 steps ($2\times$ optimal) yields 73% success. Further increases to 12 and 16 steps produces only marginal gains. These results indicate that while insufficient exploration severely limits learning, budgets beyond 2× optimal provide negligible benefits for TextWorld tasks. In contrast, **complex coding tasks benefit significantly from extended horizons.** In SWE-Gym, increasing the context window and allowed tool calls from 10 to 40 boosts success rates from 3% to 22%. Unlike simple text games, reasoning in software repositories does not saturate quickly; rather, sustained exploration allows agents to iteratively debug and refine solutions.

Table 4: Impact of exploration horizon on multi-turn performance. We compare the effect of increasing allowed exploration steps on TextWorld w2-o3-q4 tasks(Left) and allowed tool calls/response length on SWE-Gym tasks (Right).

<table>
<tr><th colspan="3">TextWorld (PPO)
*Model: Qwen-2.5-1.5B*</th><th colspan="3">SWE-Gym (GRPO)
*Model: Qwen3-8B*</th></tr>
<tr><th>Exploration Steps</th><th>Base</th><th>PPO</th><th>Tool Calls (Context)</th><th>Base</th><th>GRPO</th></tr>
<tr><td>6</td><td>0.05</td><td>$0.55_{\uparrow 0.50}$</td><td>10 (3k tokens)</td><td>0.03</td><td>$0.03_{\uparrow 0.00}$</td></tr>
<tr><td>8</td><td>0.09</td><td>$0.73_{\uparrow 0.64}$</td><td>25 (4k tokens)</td><td>0.04</td><td>$0.18_{\uparrow 0.14}$</td></tr>
<tr><td>12</td><td>0.15</td><td>$0.80_{\uparrow 0.65}$</td><td>40 (6k tokens)</td><td>0.04</td><td>$0.22_{\uparrow 0.18}$</td></tr>
<tr><td>16</td><td>0.17</td><td>$0.88_{\uparrow 0.71}$</td><td></td><td></td><td></td></tr>
</table>

## 5.2 HOW DOES MULTI-TURN RL GENERALIZE TO ENVIRONMENTS WITH DIFFERENT COMPLEXITIES?

To investigate whether multi-turn RL learns transferable skills, we evaluate cross-environment generalization. We train single-task models on w2-o3-q4, w8-o3-q4, and w2-o12-q4, plus a mixed-complexity model on equal proportions of w2-o12-q4 and w8-o3-q4, maintaining constant total RL data across conditions. Table 5 reveals that **agents trained on simpler environments achieve substantial generalization to more complex ones**, evidenced by the model trained on w2-o3-q4 that improves performance across all higher-complexity environments. Transfer is particularly strong from w8-o3-q4 (increased spatial complexity), which achieves the largest average improvements across targets—notably improving w8-o12-q4 by 48%, matching the 48% gain from training solely on w8-o12-q4. These results demonstrate that multi-turn RL acquires reusable skills like spatial exploration and object manipulation that transfer across complexity levels.

**This generalization capability extends to non-embodied domains**. As shown in Table 6, code agents trained on "Easy" tasks (based on test suite complexity) transfer effectively to harder problems, achieving gains of 4.8% on Medium and 3.6% on Hard tasks. This mirrors our findings in TextWorld, confirming that multi-turn RL acquires robust, reusable behaviors—such as error correction and state tracking—that remain effective even as the environment's complexity scales.

Table 5: Cross-environment generalization with multi-turn PPO on TextWorld tasks. All models trained with 5000 episodes/epoch (mixed-task: 2500 episodes each).

| Tasks | w2-o12-q4 | w8-o3-q4 | w8-o12-q4 | w4-o6-q8 |
|---|---|---|---|---|
| w2-o3-q4 | $0.4_{\uparrow 0.32}$ | $0.51_{\uparrow 0.44}$ | $0.27_{\uparrow 0.24}$ | $0.12_{\uparrow 0.11}$ |
| w8-o3-q4 | $0.5_{\uparrow 0.42}$ | $0.68_{\uparrow 0.61}$ | $0.51_{\uparrow 0.48}$ | $0.21_{\uparrow 0.2}$ |
| w2-o12-q4 | $0.54_{\uparrow 0.46}$ | $0.27_{\uparrow 0.2}$ | $0.27_{\uparrow 0.24}$ | $0.13_{\uparrow 0.12}$ |
| w2-o12-q4 + w8-o3-q4 | $0.41_{\uparrow 0.33}$ | $0.52_{\uparrow 0.45}$ | $0.34_{\uparrow 0.31}$ | $0.17_{\uparrow 0.16}$ |

Table 6: Cross-environment generalization with Multi-turn GRPO on SWE-Gym tasks. We evaluate transfer performance across task complexity (Metric A: lines of patch) and verification complexity (Metric B: test suite size). Diagonal elements (gray) indicate in-distribution performance.

<table>
<tr><th colspan="4">Metric A: Solution Complexity</th><th colspan="4">Metric B: Verification Complexity</th></tr>
<tr><th>Tasks</th><th>easy</th><th>medium</th><th>hard</th><th>Tasks</th><th>easy</th><th>medium</th><th>hard</th></tr>
<tr><td>easy</td><td>$0.072_{\uparrow 0.024}$</td><td>$0.048_{\uparrow 0.0}$</td><td>$0.028_{\uparrow 0.01}$</td><td>easy</td><td>$0.133_{\uparrow 0.093}$</td><td>$0.076_{\uparrow 0.048}$</td><td>$0.053_{\uparrow 0.036}$</td></tr>
<tr><td>medium</td><td>$0.161_{\uparrow 0.113}$</td><td>$0.109_{\uparrow 0.061}$</td><td>$0.055_{\uparrow 0.037}$</td><td>medium</td><td>$0.082_{\uparrow 0.042}$</td><td>$0.075_{\uparrow 0.047}$</td><td>$0.042_{\uparrow 0.025}$</td></tr>
<tr><td>hard</td><td>$0.082_{\uparrow 0.034}$</td><td>$0.091_{\uparrow 0.043}$</td><td>$0.096_{\uparrow 0.078}$</td><td>hard</td><td>$0.10_{\uparrow 0.06}$</td><td>$0.062_{\uparrow 0.034}$</td><td>$0.105_{\uparrow 0.088}$</td></tr>
</table>

## 5.3 HOW DOES MULTI-TURN RL GENERALIZE TO DIFFERENT TYPES OF TASKS WITHIN DOMAIN?

In §5.1 and §5.2, we examined how agents handle varying environment complexities within single task types. Here we address a broader question: *does our multi-turn RL recipe works for complex situated environments like ALFWorld and real-world scenarios like SWE-Gym?* Moreover, *can agents trained on task subsets generalize to full task distributions?* We investigate how training on diverse

task types affects generalization across two domains. In ALFWorld, different tasks require distinct physical skills—cleaning involves finding and placing objects, while heating requires operating appliances in specific sequences. In SWE-gym, tasks span diverse software engineering challenges from fixing pydantic issues to resolving pandas problems.

Table 7 and 8 show that **multi-turn RL successfully applies to challenging ALFworld and SWE-Gym environments**. Remarkably, **agents trained on single task types achieve decent generalization to unseen task types**, with 12% (ALFWorld) and 7% (SWE-Gym) improvements across all task types from single-type training alone. This indicates that multi-turn RL learns transferable skills even from limited task exposure. Surprisingly, multi-task training benefits seemingly unrelated tasks—agents trained on clean, heat, cool, and cook mixtures outperform single pick & place specialists by 19% on the single task itself and 21% on all-type evaluation. The results demonstrate that **multi-turn RL develops generalizable skills that transfer across diverse objectives**.

Table 7: Multi-turn PPO performance on ALFWorld unseen tasks, trained on various task mixtures.

| **Task mixture for training** | Tested on single type (PPO) | Tested on all types (PPO) |
| --- | --- | --- |
| Single type of tasks | $0.63_{\uparrow 0.19}$ | $0.59_{\uparrow 0.12}$ |
| Mixture of 4 types of tasks | $0.82_{\uparrow 0.38}$ | $0.8_{\uparrow 0.33}$ |
| Mixture of 6 types of tasks (all) | $0.76_{\uparrow 0.32}$ | $0.74_{\uparrow 0.27}$ |

Table 8: Multi-turn GRPO performance on SWE-gym tasks, trained on various task mixtures. More comments on the use of GRPO algorithm are presented in §6.2

| **Task mixture for training** | Tested on single type (GRPO) | Tested on all types (GRPO) |
| --- | --- | --- |
| Single type of tasks (getmoto) | $0.28_{\uparrow 0.19}$ | $0.11_{\uparrow 0.07}$ |
| Mixture of 5 types of tasks (all) | $0.37_{\uparrow 0.28}$ | $0.22_{\uparrow 0.18}$ |

# 6 POLICY

The choice of RL optimization algorithm and model initialization critically determines multi-turn RL performance. First, we examine the role of demonstration data. In practice, access to human demonstration data for supervised fine-tuning (SFT) may be limited. We ask: (1) *What prior model policy enables effective multi-turn RL?* This addresses whether expensive human demonstrations are necessary if agents can learn effectively from scratch. (2) *Given a fixed data budget, what is the optimal ratio of SFT to RL data?* This determines how to best allocate limited resources between demonstration collection and online learning. Second, we investigate *whether RL optimization choices significantly impact multi-turn RL training*. We compare heuristic policy gradient methods (PPO (Schulman et al., 2017), GRPO (Shao et al., 2024), Reinforce++ (Hu et al., 2025)) against unbiased methods (RLOO (Ahmadian et al., 2024)) to isolate the contributions of our multi-turn formulation from specific optimization heuristics—essential for making rigorous claims about algorithmic improvements, as demonstrated in recent work on the pitfalls of heuristic-dependent results (Oertell et al., 2025).

**Setup.** TextWorld provides gold solutions for each procedurally generated game, which we use as demonstration data for SFT, representing human multi-turn trajectories. The SFT data follows a turn-based chat format, the default template used in most instruction-following scenarios. We generate SFT data with different random seeds than RL data to prevent leakage. We train Qwen-1.5B, Qwen-7B, and Qwen-8B models using PPO, GRPO, Reinforce++, and RLOO with consistent hyperparameters across experiments. Full task specifications and training details appear in Appendices B.2 and C.2.

**Multi-turn PPO Formulation.** For optimization algorithms with advantage estimation, such as Proximal Policy Optimization (PPO) (Schulman et al., 2017), we adopt token-level credit assignment. We compute token-level values and apply to TD error as $\delta_t^i = r_t^i + \gamma V(h_t^{i+1}) - V(h_t^i)$ where $h_t^i$ is the history up to and including token $a_t^i$. Then we estimate the advantage for each token using GAE: $\hat{A}_t^i = \sum_{l=0}^{L-i}(\gamma\lambda)^l \delta_t^{i+l}$, where $L$ is the horizon (number of tokens until episode ends). Even though only `<eos>` tokens receive rewards, all preceding tokens get non-zero advantages through value bootstrapping. Therefore, the Clipped Surrogate Objective for all tokens in the trajectory can

be written as:

$$L^{CLIP}(\theta) = \mathbb{E}_{\tau \sim \pi_\theta} \left[ \sum_{t=0}^{T} \sum_{i=1}^{n_t+1} \min \left( r_t^i(\theta) \hat{A}_t^i, \text{clip}(r_t^i(\theta), 1 - \epsilon, 1 + \epsilon) \hat{A}_t^i \right) \right]$$

where the probability ratio for each token is: $r_t^i(\theta) = \dfrac{\pi_\theta(a_t^i | h_t, a_t^{<i})}{\pi_{\theta_{old}}(a_t^i | h_t, a_t^{<i})}$.

## 6.1 How does prior model policy influence multi-turn RL training?

We distinguish between establishing a prior (via SFT) and refining it through continual learning (via multi-turn online RL). This two-stage approach mirrors real-world deployment where agents learn from demonstrations before online learning. **We find that a strong SFT prior dramatically accelerates RL efficiency.** Training on golden solutions from TextWorld tasks w2-o3-q4 reveals that we can match the performance of extensive RL training with a fraction of the data. As shown in Table 9, a model initialized with just 60 demonstrations and refined over 400 RL episodes achieves 85% success, nearly matching the 88% achieved by pure RL training that requires 5000 episodes.

To determine the optimal allocation of resources, we analyze performance under a fixed budget of 1000 cost units, assuming that SFT data costs 10× more than RL episodes to reflect higher human annotation effort. **We find that while pure SFT excels at known tasks, RL is essential for robustness.** Using the full budget for SFT (100 demonstrations) shows excellent in-domain performance (95% on w2-o3-q4 tasks) but poor generalization to harder tasks (55% on w4-o6-q8 tasks). The sweet spot appears to be a hybrid configuration: 60 demonstrations followed by 400 RL episodes. This balances the behavioral grounding of SFT with the adaptability of RL, maintaining 85% in-domain success while improving generalization on the complex w4-o6-q8 task to 59%.

Table 9: Performance across SFT/RL data allocations under fixed budget. Models trained on w2-o3-q4, evaluated on w2-o3-q4 and w4-o6-q8. SFT followed by multi-turn PPO. *Results from previous experiments included for comparison.

| #SFT data | #RL data | SFT | SFT (test on w4-o6-q8) | SFT+PPO | SFT+PPO (test on w4-o6-q8) |
|---|---|---|---|---|---|
| / | 5000 | 0.17 (base) | 0.01 (base) | 0.88* | 0.12* |
| 0 | 1000 | / | / | 0.54 | 0.11 |
| 20 | 800 | 0.59 | 0.15 | 0.62 | 0.15 |
| 40 | 600 | 0.75 | 0.51 | 0.72 | 0.44 |
| 60 | 400 | 0.71 | 0.53 | 0.85 | 0.59 |
| 80 | 200 | 0.94 | 0.29 | 0.95 | 0.35 |
| 100 | 0 | 0.95 | 0.55 | / | / |

Finally, we explore whether priors transfer across distinct environments by training SFT models on ALFWorld (3553 samples) to initialize TextWorld agents, and vice versa. We find that **cross-domain priors actively harm multi-turn RL training.** Initializing with a different domain causes rapid policy collapse, likely because the specific behavioral biases and action distributions learned from the source environment conflict with the dynamics of the target environment, effectively destabilizing the policy optimization.

## 6.2 How do RL algorithms impact multi-turn RL training?

Understanding whether performance gains stem from our multi-turn formulation or specific algorithmic choices is crucial for establishing the generalizability of our approach. We compare PPO (heuristic policy gradient method with value function bootstrapping) against RLOO (unbiased policy gradient estimator), as well as recent variants like Reinforce++ and GRPO, to isolate the contributions of our token-level credit assignment from algorithmic design decisions (Oertell et al., 2025).

**Both PPO and RLOO achieve improvements over base models, showing that the performance gains stem from multi-turn formulation rather than PPO-specific heuristics.** Table 10 shows that PPO achieves 88% success on w2-o3-q4 versus RLOO's 51%. In contrast, Reinforce++ and GRPO induce negligible gains over the base model. This gap widens on harder tasks w4-o6-q8: PPO maintains 59% success while RLOO, Reinforce++, and GRPO all suffer collapse for the 1.5B model.

**Model scaling helps close the gap on simpler tasks**, and PPO maintains the best algorithm in complex reasoning. At 7B parameters, both PPO and RLOO converge on simple w2-o3-q4 tasks ($\approx$ 98%), but Reinforce++ and GRPO still lag behind (72% and 79% respectively. On the harder w4-o6-q8 tasks, PPO dominates with 72% success compared to RLOO's 47% and GRPO's 36%. These results confirm that the performance gains are not due to PPO heuristics, evidenced by RLOO's consistent improvements across tasks, but they do highlight that **PPO is significantly more sample-efficient and robust than Reinforce++ and GRPO in this domain.**

Table 10: Comparison of different RL algorithms, including PPO, RLOO, Reinforce++, and GRPO on multi-turn TextWorld tasks across model sizes.

| Task / Model | Base model | RLOO | PPO | Reinforce++ | GRPO |
|---|---|---|---|---|---|
| w2-o3-q4 / Qwen-1.5B | 0.15 | $0.51_{\uparrow 0.36}$ | $0.88_{\uparrow 0.73}$ | $0.18_{\uparrow 0.03}$ | $0.18_{\uparrow 0.03}$ |
| w4-o6-q8 / Qwen-1.5B | 0.01 | 0.0 | $0.59_{\uparrow 0.58}$ | 0.0 | $0.02_{\uparrow 0.01}$ |
| w2-o3-q4 / Qwen-7B | 0.65 | $0.97_{\uparrow 0.32}$ | $0.98_{\uparrow 0.33}$ | $0.72_{\uparrow 0.07}$ | $0.79_{\uparrow 0.14}$ |
| w4-o6-q8 / Qwen-7B | 0.28 | $0.47_{\uparrow 0.21}$ | $0.72_{\uparrow 0.44}$ | $0.33_{\uparrow 0.05}$ | $0.36_{\uparrow 0.08}$ |

We also examine GRPO in §5.3 for SWE-Gym tasks. While GRPO underperforms on the TextWorld tasks in Table 10, SWE-Gym's environment makes GRPO computationally advantageous. Therefore, we draw parallels between PPO and GRPO as biased algorithms only under sparse reward settings (only final rewards per trajectory). The improvements from GRPO in Table 8, together with PPO's results in Table 10, **show the effectiveness of biased methods in multi-turn RL.**

# 7 REWARD

Multi-turn environments typically provide sparse feedback upon task completion, creating challenges across extended sequences that can lead to slow convergence or training instability. However, some environments offer dense reward signals with partial rewards at solution milestones. We investigate how reward density, the frequency of feedback signals in trajectories, affects multi-turn RL performance and learning efficiency.

**Setup.** We investigate the impact of reward design across two distinct domains: the text-based game TextWorld and the software engineering environment SWE-Gym. For **TextWorld**, we manipulate reward density using built-in functions, comparing **sparse rewards** (task completion only) against **dense rewards** (intermediate steps). We evaluate these on the *tw-simple* tasks[2] using PPO and RLOO over 3,000 episodes. For **SWE-Gym**, we introduce a more complex evaluation of reward *source* and *granularity*. We compare **verified rewards** derived from ground-truth execution (both binary task completion and granular unit test ratios) against **model-based rewards** synthesized by judges (CodeRM-8B (Ma et al., 2025) and GPT-4.1). Given the long horizons of coding tasks, we employ GRPO for stability. Full specifications for both environments and reward configurations are detailed in Appendices B.3 and C.3.

Table 11: Performance across reward density schemes on tw-simple tasks (Qwen-7B). Parentheses show reward density as steps per reward.

| Reward density | Qwen-7B (PPO) | Qwen-7B (RLOO) |
|---|---|---|
| Sparse (10.22) | $0.41_{\uparrow 0.12}$ | $0.35_{\uparrow 0.06}$ |
| Dense 1 (2.89) | $0.29_{\uparrow 0.0}$ | $0.55_{\uparrow 0.26}$ |
| Dense 2 (1.17) | $0.58_{\uparrow 0.29}$ | $0.55_{\uparrow 0.26}$ |

## 7.1 WHAT REWARD SIGNALS ARE NEEDED FOR MULTI-TURN RL TRAINING?

**Dense rewards significantly improve multi-turn RL performance, with optimal density varying by algorithm.** As shown in Table 11, PPO benefits from more frequent feedback, with success rates jumping from 41% under sparse rewards to 58% with the densest signal (Dense 2). In contrast, RLOO maintains steady performance (55%) across both dense configurations, **suggesting its unbiased gradient estimates are less sensitive to reward density.**

---

[2]https://textworld.readthedocs.io/en/stable/textworld.challenges.simple.html

The key insight here is that reward density should be calibrated to the optimization strategy. While dense signals can accelerate convergence, their utility depends on the quality of that signal. If intermediate rewards are poorly designed, they risk introducing noise that misguides the agent.

## 7.2 VERIFIED VS. MODEL-BASED REWARDS IN COMPLEX TASKS

While TextWorld provides a controlled setting for analyzing reward density, real-world applications such as software engineering often involve complex and long-horizon reasoning where defining reward density is more challenging. To address this, we extend our analysis to the coding domain using SWE-Gym. We investigate two critical questions: (1) *Does the granularity of verified feedback matter?* and (2) *Can model-based judges substitute for verifiers?*

**The Necessity of Fine-Grained Verifier Rewards.** We first compare two implementations of execution-based feedback: a sparse *binary verified reward* (where the agent receives +1 only upon passing the full test suite) and a dense *ratio-based verified reward* (proportional to the percentage of unit tests passed). As is shown in Table 12, **fine-grained verifier rewards are essential for more complex tasks.** With sparse binary rewards, the agent achieves only a 4.2% success rate, which is a negligible improvement over the base model's 4%. However, providing dense, ratio-based feedback boosts performance to 22%. This confirms that in complex domains like software engineering, the "all-or-nothing" signal of a final test suite is too sparse to guide the policy, whereas fine-grained rew for passing individual unit tests provides the necessary gradient for learning.

**Verifiers vs. Model-Based Judges.** We also explore whether model-based rewards can serve as an effective alternative to verifiers. We employ two model-based judges capable of code generation: CodeRM-8B (Ma et al., 2025) and GPT-4.1, to generate synthetic test suites and evaluate agent's code generation progress. While model-based rewards show some improvement over the base model, they significantly underperform verified rewards. As shown in Table 12, rewards shaped by CodeRM-8B and GPT-4.1 achieve success rates of 7.2% and 9.3% respectively. Although these model-based judges provide some useful signal, they lag behind verified ratio rewards (22%) by a wide margin. Consequently, for practitioners, **the most robust recipe remains relying on execution-based unit tests rather than model-based proxies.**

Table 12: Performance of GRPO on SWE-Gym tasks with different reward signals. We compare the base model against verified rewards (binary vs. ratio) and model-based rewards (CodeRM-8B vs. GPT-4.1).

| Metric | Base Model | Verified (Binary) | Verified (Ratio) |
|---|---|---|---|
| Success Rate | 0.04 | $0.042_{\uparrow 0.002}$ | $0.22_{\uparrow 0.18}$ |
| **Model-Based Rewards** | | **Judge (CodeRM-8B)** | **Judge (GPT-4.1)** |
| Success Rate | 0.04 | $0.072_{\uparrow 0.032}$ | $0.093_{\uparrow 0.053}$ |

## 8 THE RECIPE AND CONCLUSION

Our systematic investigation across environment, policy, and reward pillars yields a practical recipe for multi-turn agentic RL, validated across TextWorld, ALFWorld, and SWE-Gym.

**Environment:** *Curriculum matters.* Start training on simpler environments or tasks with lower verification complexity. We find that agents develop reusable skills (e.g., state tracking) in easier settings that generalize effectively to complex, object-heavy, or long-horizon tasks.

**Policy:** *Prioritize stability and priors.* Biased, stabilized algorithms (PPO, GRPO) significantly outperform unbiased estimators (RLOO) and naive gradients (Reinforce++) in multi-turn settings. Furthermore, initializing with a strong SFT prior dramatically reduces RL sample complexity, offering a more efficient path to convergence than pure RL.

**Reward:** *Granularity beats approximation.* Dense rewards are essential for complex reasoning. However, our results show that **verified execution feedback** (e.g., unit test pass rates) is far superior to model-based surrogates. Practitioners should prioritize engineering granular, execution-based signals over training reward models.

By systematically identifying this recipe: **Curriculum + Stabilized Policy + Verified Dense Reward**, we provide the empirical foundation and practical guidelines for developing truly autonomous agents capable of extended real-world interaction.

REPRODUCIBILITY STATEMENT

We are committed to ensuring the reproducibility of our experimental results. To this end, we provide comprehensive details about our experimental setup and will release all necessary resources for replication:

**Code and Framework**: We will open-source our complete multi-turn RL framework built on veRL, including environment integrations for TextWorld, ALFWorld, and SWE-Gym. All training scripts, evaluation pipelines, and data generation procedures will be included in the repository upon publication.

**Experimental Details**: Full hyperparameters for all experiments are provided in Appendices, including learning rates, KL penalties, batch sizes, exploration budgets, and temperature settings for each environment and model configuration. We specify exact model versions (Qwen2.5-1.5B-Instruct, Qwen2.5-7B-Instruct, Qwen3-8B), training epochs, and convergence criteria. Task selection procedures, including random seeds for procedural generation in TextWorld and specific task splits for ALFWorld and SWE-Gym, are also documented.

**Data and Models:** We detail our data generation process, including the creation of SFT demonstrations from TextWorld gold solutions and the sampling strategy for SWE-Gym tasks. Model weights for key experimental configurations will be released. For environments requiring specific versions, we provide exact package versions and installation instructions.

**Computational Requirements:** All experiments were conducted on NVIDIA H100 GPUs. We report approximate training times and computational resources required for each experimental configuration, enabling researchers to estimate reproduction costs. The framework supports distributed training for efficient replication at scale.

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

## A    HYPERPARAMETER TUNING FOR MULTI-TURN RL

To establish a robust training recipe for multi-turn RL, we conducted systematic hyperparameter sweeps training Qwen-1.5B on TextWorld w2-o3-q4 tasks using PPO. Following the setup in §4, we train on 5,000 episodes and evaluate on 100 held-out episodes. Each experiment runs for 30 epochs (approximately 575 steps) to assess early training stability, measuring task success rate throughout training.

Figure 2 reveals substantial performance variation across hyperparameter configurations. Higher KL coefficients ($> 0.001$) yield more stable training curves. Comparing configurations shows that gamma values below 1.0 impair performance (blue vs. pink curves), while rollout temperature critically affects outcomes—optimal performance occurs between 0.7 and 1.0 (comparing light green, pink, and dark green curves). Higher learning rates for both actor and critic networks improve training efficiency and final performance (brown vs. light green curves).

Table 13 confirms the optimal configuration: KL coefficient of 0.01, temperature of 0.7, actor learning rate of 1e-6, critic learning rate of 1e-5, and gamma of 1.0. These settings provide both training stability and strong final performance, forming the basis for our experiments across all benchmarks.

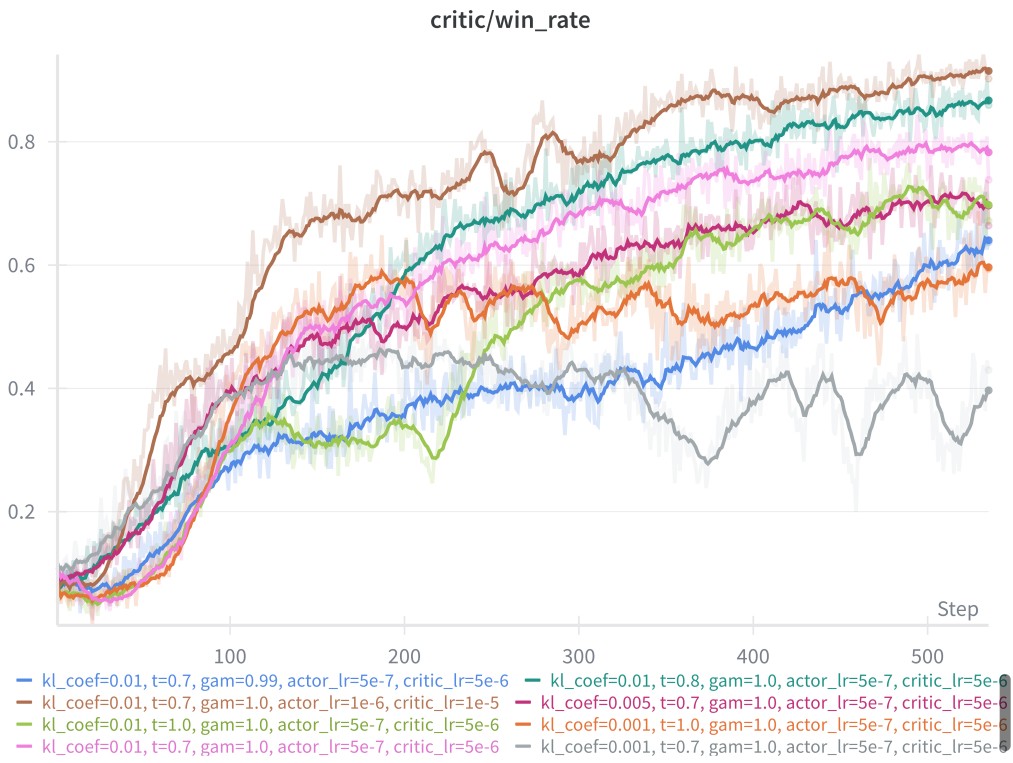

Figure 2: Training curves for Qwen-1.5B on TextWorld w2-o3-q4 with varying hyperparameters. Parameters shown: KL coefficient (kl_coef), rollout temperature (t), actor/critic learning rates, and discount factor (gam).

Table 13: Success rates at epoch 30 on TextWorld w2-o3-q4 test set, sorted by performance. Only top-performing configurations shown; excluded configurations performed substantially worse.

| Ranking | kl_coef | temperature | actor_lr | critic_lr | gamma | win rate |
|---------|---------|-------------|----------|-----------|-------|----------|
| 1 | **0.01** | **0.7** | **1e-6** | **1e-5** | **1.0** | **0.9** |
| 2 | 0.01 | 0.8 | 5e-7 | 5e-6 | 1.0 | 0.86 |
| 3 | 0.01 | 0.7 | 5e-7 | 5e-6 | 1.0 | 0.78 |
| 4 | 0.01 | 1.0 | 5e-7 | 5e-6 | 1.0 | 0.71 |
| 5 | 0.005 | 0.7 | 5e-7 | 5e-6 | 1.0 | 0.66 |
| 6 | 0.01 | 0.7 | 5e-7 | 5e-6 | 0.99 | 0.66 |
| 7 | 0.001 | 1.0 | 5e-7 | 5e-6 | 1.0 | 0.57 |
| 8 | 0.001 | 0.7 | 5e-7 | 5e-6 | 1.0 | 0.43 |

## B    DETAILS OF TASKS AND ENVIRONMENTS

We show an example of the TextWorld w2-o3-q4 task in Figure 3. The ALFWorld tasks are built on TextWorld and they are similar in format, but the tasks in ALFWorld rely more on agents' embodied understanding. An example from ALFWorld heat & place task is presented in Figure 4.

We extract SWE-Gym tasks from their huggingface repository (https://huggingface.co/datasets/SWE-Gym/SWE-Gym. An example from SWE-Gym getmoto task is presented in Figure 5.

### B.1    ADDITIONAL TASK DETAILS FOR §5

For environment complexity, we procedurally generated 5000 RL episodes for w2-o3-q4, w8-o3-q4, w2-o12-q4, and w4-o6-q8 respectively using TextWorld's `tw-make`, using random seeds ranging from 30001 to 35000.

For task complexity using ALFWorld, we make sure the number of data for each training mixture is the same, which is 1000. Here is a brief data statistics: (1) single-type tasks contain 1000 pick & place tasks, randomly sampled from 3553 train data; (2) mixed-type (4 types) contain 250 heat, cool, cook, and examine tasks respectively; (3) mixed-type (all types), randomly sampled 1000 any type of tasks from 3553 train data.

For task complexity using SWE-Gym, we make sure the number of data for each training mixture is the same, which is 90. Here is a brief data statistics: (1) single-type tasks contain 90 getmoto tasks, randomly sampled from the SWE-Gym huggingface repo; (2) mixed-type (5 types) contain 18 getmoto, pydantic, mypy, pandas, and interactive_dvc tasks respectively.

### B.2    ADDITIONAL TASK DETAILS FOR §6

We collect 0/20/40/60/80/100 SFT data from the gold trajectories from TextWorld games. The seeds range from 40001 to 40101, different from the RL data.

### B.3    ADDITIONAL TASK DETAILS FOR §7

The tasks are procedurally generated using TextWorld's built-in `tw-simple` function, which has a dense reward calculator for each step. We utilize TextWorld's built-in reward function to assign rewards per step. We collect 3000 episodes with sparse, dense 1, and dense 2 density levels, respectively.

## C    DETAILS OF TRAINING AND EVALUATION

We train Qwen-1.5B, Qwen-7B, and Qwen-8B on 8 NVIDIA H100 GPUs.

You are an expert TextWorld game solver. Your goal is to generate the best next action that will lead to winning the game.

End your output sequence with an action starting with a verb. Example: open box.

Here is how to win the game:
**I hope you're ready to go into rooms and interact with objects, because you've just entered TextWorld! Here is your task for today. Your first objective is to take the type J keycard from the case in the studio. After that, make it so that the type J chest is unlocked with the type J keycard. Then, ensure that the type J chest is open. Then, take the Quote of the Day Calendar from the type J chest in the studio. And if you do that, you're the winner!**

Here is your interactions so far:
**current state:** You are now in the studio.
You find yourself in a studio. An ordinary one. The room is well lit.
You see a locked type J chest. You lean against the wall, inadvertently pressing a secret button. The wall opens up to reveal a case. The case contains a type J keycard.
You need an unblocked exit? You should try going north.
You are carrying nothing.

**your action:** take type J keycard from case

**current state:** You are now in the studio.
You take the type J keycard from the case.
You are carrying: a type J keycard.

**your action:** unlock type J chest with type J keycard

**current state:** You are now in the studio.
You unlock the type J chest.
You are carrying: a type J keycard.

**your action:** open type J chest

**current state:** You are now in the studio.
You open the type J chest, revealing a Quote of the Day Calendar.
You are carrying: a type J keycard.

**your action:** take Quote of the Day Calendar from type J chest

**Final reward: 1**

Figure 3: Textworld w2-o3-q4 task example. The text in gray are the prompts. The bold text is the objective. The text in blue are the observations and the text in orange are the actions.

You are an expert TextWorld game solver. Your goal is to generate the best next action that will lead to winning the game.

End your output sequence with an action starting with a verb. Example: open box.

Here is how to win the game:
**heat some apple and put it in fridge.**

Here is your interactions so far:
**current state:** -= Welcome to TextWorld, ALFRED! =-\n\nYou are in the middle of a room. Looking quickly around you, you see a cabinet 6, a cabinet 5, a cabinet 4, a cabinet 3, a cabinet 2, a cabinet 1, a coffeemachine 1, a countertop 3, a countertop 2, a countertop 1, a drawer 3, a drawer 2, a drawer 1, a fridge 1, a garbagecan 1, a microwave 1, a shelf 3, a shelf 2, a shelf 1, a sinkbasin 1, a stoveburner 4, a stoveburner 3, a stoveburner 2, a stoveburner 1, and a toaster 1.\n\nYour task is to: heat some apple and put it in fridge.

**your action:** go to garbagecan 1

**current state:** You arrive at garbagecan 1. On the garbagecan 1, you see a apple 2, and a soapbottle 3.

**your action:** take apple 2 from garbagecan 1

**current state:** You pick up the apple 2 from the garbagecan 1.

**your action:** go to microwave 1

......

Figure 4: ALFWorld heat & place task example. The text in gray are the prompts. The bold text is the objective. The text in blue are the observations and the text in orange are the actions. We only provide part of the task trajectory because it is very long and is similar to the TextWorld example.

You are a helpful assistant that can interact multiple times with a computer shell to solve programming tasks......
Here is your interactions so far:

**current state:**
<uploaded_files></uploaded_files>
<pr_description>Consider the following PR description: ......
</pr_description>

**your action:**
<think>Okay, let's tackle this problem. The user reported that when using moto's mock_ec2, the describe_security_group_rules method isn't respecting the filter......
</think>
**find . -type f -exec grep -l \"describe_security_group_rules\" {} \\;**

**current state:**
./moto/ec2/models/security_groups.py
/moto/ec2/responses/security_groups.py
/CHANGELOG.md
./IMPLEMENTATION_COVERAGE.md
......

**your action:**
<think>The `describe_security_group_rules` method is likely implemented in `moto/ec2/models/security_groups.py` based on the grep result. I'll examine this file to understand how......
</think>
**grep -r \"describe_security_group_rules\" moto/ec2/models/**

......

**your action:**
<think>......</think>
**cat <<'EOF' > test_moto_security_group.py\nimport boto3 ... (rest of the patch)**

**current state:**
Your last command ran successfully and did not produce any output.

**your action:**
**submit**

**Final reward: 1**

Figure 5: SWE-Gym getmoto task example. The text in gray are the prompts. The bold text is the objective. The text in blue are the observations and the text in orange are the actions.

For PPO, by default, we use an actor learning rate of 5e-7, a critic learning rate of 5e-6, a clip ratio of 0.2, a discount factor $\gamma$ of 1.0, a KL penalty coefficient of 0.001, batch size of 256, and a zero entropy regularization for both Qwen-1.5B and Qwen-7B models.

For GRPO, by default, we use an actor learning rate of 1e-6, KL loss coefficient of 0.001, GRPO sampling number of 4, and batch size of 16 for Qwen3-8B model.

For RLOO, by default, we use an actor learning rate of 1e-6, a KL penalty coefficient of 0.001, and batch size of 256 for both Qwen-1.5B and Qwen-7B models.

For SFT, by default, we use learning rate of 1e-6.

## C.1 Additional training details for §5

For TextWorld experiments, we train 150 epochs and evaluate on 100 held-out test sets (with seeds 1 to 100) at epoch 150.

For ALFWorld experiments, given that the base model has zero accuracy on the tasks, we do one epoch of SFT on 300 data, with different seeds from RL data. After the SFT initialization, we train 90 epochs and evaluate on 100 held-out test sets at epoch 90.

For SWE-Gym experiments, we train 15 epochs and evaluate on 25 held-out test sets at epoch 15.

## C.2 Additional training details for §6

For SFT, we do exactly one epoch, ensuring there is only one pass on the demonstration data to avoid overfitting. We train RL on top of the SFT for another 100 epochs.

For RLOO and PPO comparison, we train 150 epochs for each, similar to Appendix C.1.

## C.3 Additional training details for §7

For PPO and RLOO training for dense rewards, we train 150 epochs for each and test on 100 held-out data at epoch 150.

## D Training curves for PPO versus RLOO policies on TextWorld tasks

## E Use of LLMs

We used large language models (specifically Claude) exclusively for text polishing and grammar checking during the preparation of this manuscript. The LLM was used to improve clarity, fix grammatical errors, and enhance the conciseness of our writing. All experimental design, analysis, interpretations, and scientific conclusions are entirely our own original work. No LLMs were used for generating experimental results, creating figures or tables, or producing any technical content. The core research ideas, methodology, and findings presented in this paper are the product of human authorship.

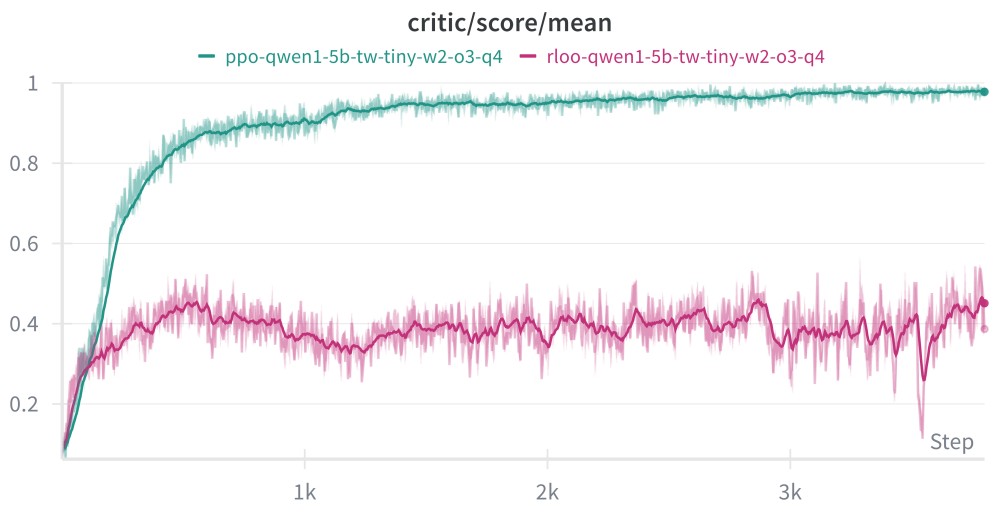

Figure 6: The training reward curve for Qwen-2.5-1.5B model trained using PPO and RLOO on TextWorld w2-o3-q4 tasks.

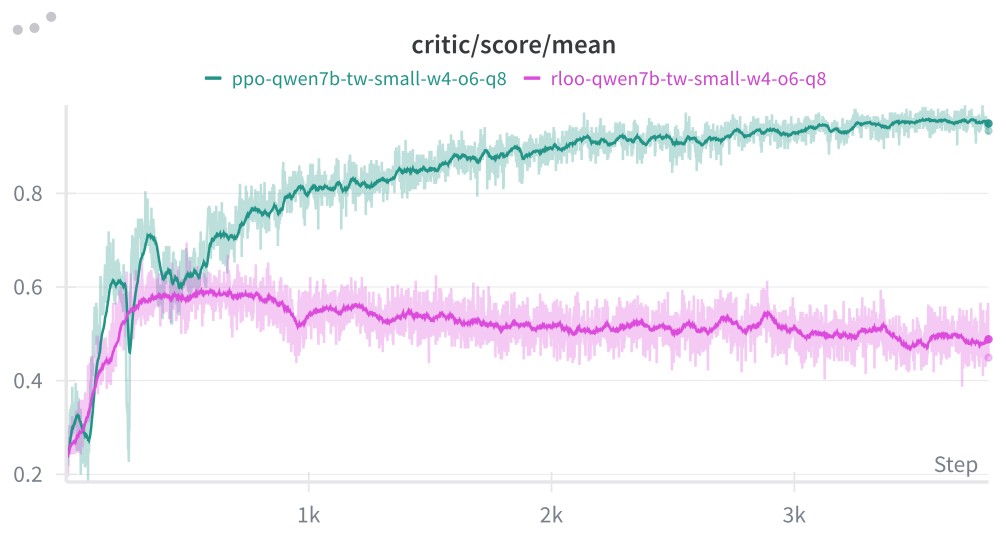

Figure 7: The training reward curve for Qwen-2.5-7B model trained using PPO and RLOO on TextWorld w4-o6-q8 tasks.

