# OpenReview forum: "A Practitioner's Guide to Multi-turn Agentic Reinforcement Learning"
_ICLR.cc/2026/Conference — Submitted to ICLR 2026_

### Official Review · Reviewer_iMhR · 2025-10-27

**Soundness:** 3
**Presentation:** 3
**Contribution:** 2
**Rating:** 4
**Confidence:** 4

**Summary:**

This paper presents a systematic empirical study on training large language models as multi-turn agents via reinforcement learning. The authors decompose the design space into three pillars (environment, reward, policy) and explore how factors affect training performance. Experiments are conducted with various SFT:RL data ratios. The work distills a set of practical recipes for effective multi-turn RL training.

**Strengths:**

1. This work provides a comprehensive empirical analysis of design choices across multiple benchmarks and domains.
2. The study bridges fragmented literature and delivers practical guidelines useful for practitioners in agentic RL.
3. Writing and organization are clear with details.

**Weaknesses:**

1. The work mostly summarizes founded intuitions without introducing fundamentally new algorithms or theories. It's a good work that delievers engineering experiences, but hard to bring new findings for a research submission.
2. The authors are recommended to conduct deeper theoretical or practical analysis to one of the observation that may deliever good findings.
3. The authors tested on severl domains of textual environments. The reviewer is curious whether the findings can be generalized to multimodal or real-world interactive settings?

**Questions:**

This work is valuable as a practical guide and empirical survey for researchers building multi-turn RL agents. However, the lack of methodological novelty and limited depth of analysis weaken its contribution.

---

> ### Author Response · Authors · 2025-12-03
> **Response to Reviewer iMhR**
>
> ***Weakness 1***: *“The work mostly summarizes founded intuitions without introducing fundamentally new algorithms or theories. It's a good work that delievers engineering experiences, but hard to bring new findings for a research submission.”*
>
> ***Response 1***: We respectfully argue that in the rapidly evolving field of Agentic RL, distinguishing “intuitions” from "empirical facts" is a critical research contribution.
>
> For example, if you only look at simple Math/Code tasks), you may derive the intuition that "unbiased estimators like RLOO are sufficient to solve agentic tasks”. However, our work empirically disproves this intuition for long-horizon agents, showing that RLOO fails due to variance accumulation in multi-turn settings (Paper section 6). We believe this kind of engineering-heavy empirical analysis is essential to prevent the community from building on shaky assumptions.
>
> ----------------------------------------------------------------------
>
> ***Weakness 2***: *“The authors tested on severl domains of textual environments. The reviewer is curious whether the findings can be generalized to multimodal or real-world interactive settings?”*
>
> ***Response 2***: Yes. Our framework is formulated for general POMDPs, where the “observation” can be text, code, or pixels. The core challenges we analyze can be applied to multimodal and real-world interactive settings. In fact, one of our tasks, SWE-Gym, is a real-world software engineering task.
>
> -------------------------------------------------------
>
> ***Questions:***
> *“This work is valuable as a practical guide and empirical survey for researchers building multi-turn RL agents. However, the lack of methodological novelty and limited depth of analysis weaken its contribution.”*
>
> ***Answer***: See Response 1 to Weakness 1

---

### Official Review · Reviewer_WXhe · 2025-11-01

**Soundness:** 2
**Presentation:** 3
**Contribution:** 2
**Rating:** 4
**Confidence:** 3

**Summary:**

This paper tackles a timely and important problem: creating a "recipe" for training multi-turn RL agents. The "three-pillar" (Environment, Policy, Reward) framework is a clear and useful way to structure the problem. The "Environment" pillar (Section 5) in particular is high quality, with solid empirical work.

However, the paper's core contributions to the "Policy" and "Reward" pillars are severely undermined by a failure to engage with (or in some cases, even cite) highly relevant, contemporaneous work. The "Reward" pillar's analysis is outdated, as it misses the entire SOTA on step-wise credit assignment. The "Policy" pillar's main experiments (both on SFT:RL ratios and algorithm comparisons) are not novel, as other papers directly covered them. The claim to a "systematic framework" is also scooped by prior work.

**Strengths:**

This paper tackles an important and, frankly, very timely problem. The idea of creating a "practitioner's guide" or "recipe" for training multi-turn RL agents is definitely needed, given the many fragmented approaches.

I found the paper's strengths to be in its structure, its empirical rigor in one of its pillars, and its timeliness.

1. Clarity: The "three-pillar" framework (Environment, Policy, Reward) is a very clear and effective mental model. It’s a great way to structure the problem space, especially for newcomers or practitioners, which seems to be the target audience. It makes the paper easy to read and follow.
2. Quality (The "Environment" Pillar): The analysis in Section 5 (Environment) is, in my opinion, the strongest part of this paper. The authors didn't just use standard benchmarks; they methodically controlled for different axes of complexity (spatial, object, quest length). The finding that "object complexity proves more challenging than spatial complexity" is a concrete, non-obvious, and valuable insight for anyone designing a training curriculum. The hyperparameter tuning detailed in Appendix A also shows solid, careful empirical work.
3. Originality (of Motivation): I want to give the authors credit for having their finger on the pulse of the community. The motivation for Section 6.2—comparing biased vs. unbiased algorithms—is excellent. They correctly identify the very recent and important debate sparked by "Heuristics Considered Harmful: RL With Random Rewards Should Not Make LLMs Reason" (Oertell et al., 2025). Grounding their empirical comparison of PPO and RLOO in this very current theoretical dispute is a good piece of scholarship.
4. Significance: The effort to validate the "recipe" across three very different domains (TextWorld, ALFWorld, and SWE-Gym) is commendable. This cross-domain validation is crucial if the paper is to live up to its "practitioner's guide" title, and it increases the potential impact of the findings.

**Weaknesses:**

Despite the strengths, this paper suffers from several major, and I mean major, weaknesses related to novelty and its literature review. The research area of LLM agents is moving incredibly fast, and unfortunately, this paper has been "scoped" by several other key papers published in the first half of 2025.

These omissions are so significant that they call into question the paper's core contributions.

1. Major Weakness 1: The "Reward" Pillar (Section 7) is Fundamentally Outdated. Frankly, this section is the paper's most significant weakness. The analysis only compares sparse rewards to the built-in dense rewards from TextWorld. This might have been an interesting question in 2023, but by 2025, the SOTA has moved far beyond this. We know dense rewards help; the critical research question now is how to generate fine-grained, step-wise credit when built-in rewards don't exist.

   The paper completely fails to cite or compare against the entire wave of 2024-2025 research on this exact topic. This includes, but is not limited to:

   - StepAgent (Deng et al., 2024), which generates step-wise rewards by comparing agent and expert policies.
   - GiGPO (Feng et al., 2025), which introduced a "Group-in-Group" optimization for fine-grained, step-level credit assignment.
   - SPA-RL (Choudhury et al., 2025), which uses a "progress estimator" to decompose the final reward into step-wise contributions.
   - MT-GRPO (Zeng et al., 2025), which also introduced fine-grained turn-level advantage estimation.

   For a "practitioner's guide" to ignore all of these SOTA reward mechanisms—the very techniques practitioners are trying to implement—is a fatal flaw. The "recipe" in this pillar is incomplete and already obsolete.

2. Major Weakness 2: The "Policy" Pillar (Section 6) Lacks Novelty. This is the second major blow to the paper's contribution. The two core experimental questions asked in this section were both answered by other papers.

   - On the SFT:RL Ratio (Sec 6.1): The authors explore the "optimal SFT:RL ratio" by testing about 6 configurations (Table 7). However, arXiv:2507.04103v1 is a large-scale study that does exactly this, running 1,370 training configurations to find the "optimal compute allocation and hyperparameter mix". This paper's small-scale analysis is completely overshadowed and can only be considered a minor validation, not a novel contribution.
   - On Algorithm Comparison (Sec 6.2): The paper's comparison of PPO, GRPO, and RLOO is also not novel. arXiv:2507.14897v1 explicitly states that it "evaluate[s] four RL algorithms:... PPO,... GRPO, and RLOO". This is a direct experimental overlap. The authors' main contribution in this section has been done concurrently (or, from the perspective of this submission, previously).

3. Major Weakness 3: The "Systematic Framework" Claim is Incorrect. The paper's abstract and introduction claim to provide the first "systematic formulation" for this problem. This is simply not true.

   - RAGEN (Wang et al., 2025), explicitly introduced the StarPO framework and the RAGEN system as a "research infrastructure for systematic analysis" of multi-turn RL agents.
   - The authors do cite RAGEN, but they dismiss it in a single line ("rely on sparse terminal rewards" - line 106), which isn't even fully accurate, as the RAGEN paper discusses the need for fine-grained rewards. The authors fail to engage with this prior work and do not clearly differentiate their "three-pillar" framework conceptually from the RAGEN/StarPO framework.

4. Major Weakness 4: No SOTA Baseline Comparisons. For a paper that claims to be a "guide," it does a poor job of benchmarking its own "recipe." In Table 5 (ALFWorld) and Table 6 (SWE-Gym), the results are presented in a vacuum.

   - Where is the comparison to the SOTA on ALFWorld?
   - SPA-RL (Choudhury et al., 2025) reported 79.1% success on unseen ALFWorld tasks. This paper's best result in Table 5 is 74% (SFT+PPO).
   - GiGPO (Feng et al., 2025) also reported a >12% gain over GRPO on ALFWorld.
   - The paper presents a recipe that is sub-SOTA and doesn't even acknowledge, let alone discuss, why. This is a critical omission.

**Questions:**

1. (Reward Pillar): Your analysis in Section 7 is limited to sparse vs. built-in dense rewards. Can you please justify why you completely omitted any discussion or comparison with the entire line of work on learned or decomposed step-wise rewards, such as StepAgent (Deng et al., 2024), GiGPO (Feng et al., 2025), and SPA-RL (Choudhury et al., 2025)? How can this be a "practitioner's guide" if it ignores the SOTA reward-design techniques?
2. (Policy Pillar - Algos): This is a direct question about novelty. Given that arXiv:2507.14897v1 already provides an empirical comparison of PPO, GRPO, and RLOO, what new insight does your Section 6.2 analysis provide that is not already present in that work? To be very clear, you need to explain what your contribution is over and above that paper.
3. (Policy Pillar - Ratios): Similarly, how do you position your findings on the SFT:RL ratio (Sec 6.1) against the large-scale, 1,370-run study in arXiv:2507.04103v1? Are your findings (from ~6 runs) simply a small-scale confirmation, or do you arrive at a different conclusion?
4. (Framework Claim): Can you please elaborate in detail on the conceptual novelty of your "three-pillar" framework compared to the "systematic analysis" framework proposed by RAGEN (Wang et al., 2025)? The current dismissal in your related work section is insufficient.

---

> ### Author Response · Authors · 2025-12-03
> **Response to Reviewer WXhe - 1**
>
> ***Major Weakness 1***: *“The "Reward" Pillar (Section 7) is Fundamentally Outdated. Frankly, this section is the paper's most significant weakness. The analysis only compares sparse rewards to the built-in dense rewards from TextWorld. This might have been an interesting question in 2023, but by 2025, the SOTA has moved far beyond this. We know dense rewards help; the critical research question now is how to generate fine-grained, step-wise credit when built-in rewards don't exist. The paper completely fails to cite or compare against the entire wave of 2024-2025 research on this exact topic. This includes, but is not limited to:
> StepAgent (Deng et al., 2024), which generates step-wise rewards by comparing agent and expert policies.
> GiGPO (Feng et al., 2025), which introduced a "Group-in-Group" optimization for fine-grained, step-level credit assignment.
> SPA-RL (Choudhury et al., 2025), which uses a "progress estimator" to decompose the final reward into step-wise contributions. [May 27, 2025]
> MT-GRPO (Zeng et al., 2025), which also introduced fine-grained turn-level advantage estimation.
> For a "practitioner's guide" to ignore all of these SOTA reward mechanisms—the very techniques practitioners are trying to implement—is a fatal flaw. The "recipe" in this pillar is incomplete and already obsolete.”*
>
> ***Response 1***:
> We thank the reviewer for highlighting these ***concurrent*** works. We agree that the frontier has shifted towards deriving fine-grained rewards when ground truth is absent. We will cite and discuss these papers in the related work section of the final version.
>
> First, we respectfully clarify that our goal is not to propose a new reward assignment method; instead, we try to answer **“how and why reward density affects training stability across RL algorithms”**. In this context, the “built-in” rewards of TextWorld tasks serve as an **oracle upper bound**, representing the ideal signal that methods like StepAgent and GiGPO aim to approximate.
>
> Second, we respectfully disagree that the exclusion of the listed papers makes our paper obsolete. We make sure our work is **reproducible and accessible**. At the time of writing (and ***concurrently***):
> - StepAgent: No official code released
> - SPA-RL: No process reward model checkpoints released
> - MT-GRPO: Evaluated primarily on short-horizon tasks (single/two-turn), limiting its applicability to the multi-turn agents we study.
>
> Because these methods are inaccessible to practitioners, we could not include them as baselines (except for GiGPO, which we will discuss in ***Response 4*** in detail). Instead, we focused on **practical and reproducible** strategies for synthesizing dense rewards that practitioners can implement today:
> - **Heuristics-based reward** that is derived from the environment (through exploration bonuses, subgoal completions, etc.). This is already included in our paper Section 7.1.
> - **Model-based reward** that is provided by reward model trained for similar tasks, or SOTA LLM-as-a-judge. We include this in the **new Section 7.2** in new PDF.
>
> To address your core concern that we ignored the fine-grained reward in complex tasks, we add the following experiments:
>
> **Experiment 1. Analyzing dense verified reward in complex tasks (SWE-Gym):** We move beyond TextWorld to the coding environment. We compare two reproducible reward strategies:
> 1. **Sparse (binary verified reward)**: Agent receives +1 at the end only if it passes the test suite.
> 2. **Dense (ratio-based verified reward)**: Agent receives reward proportional to the percentage of unit tests passed so far.
>
> As shown in the following table (same as ***new Table 12*** in new PDF), in SWE-Gym tasks, sparse reward achieves 4.2% performance while dense reward achieves 22%. The improvement empirically shows that fine-grained credit assignment is critical for complex tasks. Importantly, both binary and ratio-based verified reward is a practical dense reward that any engineer can implement with a test suite in software coding tasks.
>
> ***Table WXhe.1: Performance of Qwen3-8B model trained using multi-turn GRPO on SWE-Gym tasks with different reward types, including verified (binary, ratio-based) and model-based (CodeRM-8B as judge, GPT-4.1 as judge). * means reporting numbers from our paper.***
> | Tasks / rewards | Base model\* | Verified (binary) | Verified (ratio) | Model judge (CodeRM-8B) | Model judge (GPT-4.1) |
> | :---- | :---- | :---- | :---- | :---- | :---- |
> | SWE-Gym (GRPO) | 0.04\* | 0.042 (+0.002) | 0.22 (+0.18) | 0.072 \+ (0.032) | 0.093 \+ (0.053) |
>
> ***(to be continued...)***

---

> ### Author Response · Authors · 2025-12-03
> **Response to Reviewer WXhe - 2**
>
> ***(continue from above...)***
>
>
> **Experiment 2. Exploring model-based reward shaping (SWE-Gym)**: We further investigate synthesizing reward using model-based judges to generate test suites and evaluate agents’ progress. We use two types of model-based judges: **CodeRM-8B**[1], an open-source reward model trained on the OpenCodeInterpreter dataset[2]; and **GPT-4.1**. They are used to generate unit tests based on the code problem statement and the current solution patch the agent generated. The dense reward is the passing ratio of the unit tests during the interaction.
>
> Furthermore, to enable stable training of long-horizon SWE-Gym tasks with dense reward, we implement multi-turn GRPO[3], the RL method used by Meta’s Code World Model that is released after our paper is submitted. The result is shown in the above table (same as ***new Table 12*** in new PDF), we see that both model-based rewards achieve some improvement (3.2% and 5.3%, respectively) upon the base model, with the GPT-4.1 judge performing slightly better than CodeRM-8B. An interesting finding is that the model-based reward underperforms verifier reward (18%), suggesting that verifier reward is essential for coding tasks.
>
> **In summary, our new experiments confirm that our “standard recipe”: dense verified rewards + PPO/GRPO remains the robust and reproducible approach for practitioners.**
>
>
> References:
>
> [1] Ma, Zeyao, et al. "Dynamic scaling of unit tests for code reward modeling." arXiv preprint arXiv:2501.01054 (2025).
>
> [2] Zheng, Tianyu, et al. "Opencodeinterpreter: Integrating code generation with execution and refinement." arXiv preprint arXiv:2402.14658 (2024).
>
> [3] Copet, Jade, et al. "CWM: An Open-Weights LLM for Research on Code Generation with World Models." arXiv preprint arXiv:2510.02387 (2025).

---

> > ### Author Response · Authors · 2025-12-03
> > **Response to Reviewer WXhe - 3**
> >
> > ***Major Weakness 2***: *“The "Policy" Pillar (Section 6) Lacks Novelty. This is the second major blow to the paper's contribution. The two core experimental questions asked in this section were both answered by other papers.*
> >
> > *On the SFT:RL Ratio (Sec 6.1): The authors explore the "optimal SFT:RL ratio" by testing about 6 configurations (Table 7). However, arXiv:2507.04103v1 is a large-scale study that does exactly this, running 1,370 training configurations to find the "optimal compute allocation and hyperparameter mix". This paper's small-scale analysis is completely overshadowed and can only be considered a minor validation, not a novel contribution.*
> >
> > *On Algorithm Comparison (Sec 6.2): The paper's comparison of PPO, GRPO, and RLOO is also not novel. arXiv:2507.14897v1 explicitly states that it "evaluate[s] four RL algorithms:... PPO,... GRPO, and RLOO". This is a direct experimental overlap. The authors' main contribution in this section has been done concurrently (or, from the perspective of this submission, previously).”*
> >
> > ***Response 2***: We thank the reviewer for citing these ***concurrent works (both in July 2025)***. While we acknowledge some similarities in topics, we respectfully demonstrate that our work investigates **distinct research questions** with different practical implications for practitioners.
> >
> > 1. **SFT:RL ratio** (Section 6.1): The cited concurrent work is a valuable large-scale study for SFT:RL compute allocation that covers generalization to held-out tasks. However, our study provides two distinct contributions:
> >     - **Cost constraints vs. compute scaling**: The cited work focuses on SFT:RL scaling laws for compute allocation (FLOPs). Our paper focuses on the cost constraints of SFT:RL data. We model the scenario where expert SFT/demonstration data is significantly more expensive (10x) than RL experience. We identify a specific cost-performance optimization problem, where a mix of SFT+RL training paradigms achieves better performance under fixed annotation budgets. This framing is absent in prior work and provides guidance for teams with limited data budgets, with a focus deviated from compute budgets.
> >     - **Controllable SFT+ RL easy-to-hard generalization**: The cited work assesses SFT+RL generalization to random unseen tasks, which is covered by our analysis as well. More importantly, our work focuses on easy-to-hard generalization defined by our environment complexity metric. We explicitly analyze how models trained on low-complexity environments (TextWorld w2-o3-q4) transfer to high-complexity ones (w4-o6-q8). This provides specific guidance on designing training curricula (for example, “train on easy tasks that are quicker and cost less, and deploy on harder tasks”), which is a distinct objective from generalization to held-out distribution.
> > 2. **RL algorithm comparison**: The cited concurrent work evaluates different RL algorithms on Code Interpreter Math tasks (see their Figure 3), which are short-horizon math problems often solvable in 1-2 turns of code generation). It differs from the multi-turn agentic tasks we study (TextWorld/ALFWorld with 10-20 turns, and SWE-Gym with 20+ turns). Plus, we use 7B, the industrial accepted smallest parameter size which you can draw results from; while they use 3B models without parameter sweeping.
> >
> > To make our analysis more convincing, we add two more RL algorithms **(Reinforce++, GRPO)** on different TextWorld tasks **(easy and harder sets)**, across different model sizes **(1.5B and 7B)**, shown in the following tables (same as **new Table 10** in new PDF). Therefore, we provide a more up-to-date and relevant guide on RL algorithms for building long-horizon agents than previous math-focused study.
> >
> > ***Table WXhe.2:  Performance of the Qwen2.5-1.5B model and Qwen2.5-7B model trained using different multi-turn RL algorithms, on TextWorld w2-o3-q4 tasks. * means reporting numbers from our paper.***
> > | Model/Algo | Base\* | \+RLOO\* | \+Reinforce++ | \+PPO\* | \+GRPO |
> > | :---- | :---- | :---- | :---- | :---- | :---- |
> > | Qwen-1.5B | 0.15\* | 0.51\* | 0.18 | 0.88\* | 0.18 |
> > | Qwen-7B | 0.65\* | 0.97\* | 0.72 | 0.98\* | 0.79 |
> >
> > ***Table WXhe.3: Performance of the the Qwen2.5-1.5B model and Qwen2.5-7B model trained using different multi-turn RL algorithms, on TextWorld w4-o6-q8 tasks. * means reporting numbers from our paper.***
> > | Model/Algo | Base\* | \+RLOO\* | \+Reinforce++ | \+PPO\* | \+GRPO |
> > | :---- | :---- | :---- | :---- | :---- | :---- |
> > | Qwen-1.5B | 0.01\* | 0.0\* | 0.0 | 0.59\* | 0.02 |
> > | Qwen-7B | 0.28\* | 0.47\* | 0.33 | 0.72\* | 0.36 |
> >
> > **In summary, our work does not repeat the cited studies in that: 1) we test SFT:RL paradigm under cost constraints and controllable easy-to-hard generalization and 2) we provide analysis across RL algorithms (PPO, GRPO, RLOO, Reinforce++) in long-horizon and true multi-turn agentic tasks.**

---

> > > ### Author Response · Authors · 2025-12-03
> > > **Response to Reviewer WXhe - 4**
> > >
> > > ***Major Weakness 3***: *“The "Systematic Framework" Claim is Incorrect. The paper's abstract and introduction claim to provide the first "systematic formulation" for this problem. This is simply not true.*
> > >
> > > *RAGEN (Wang et al., 2025), explicitly introduced the StarPO framework and the RAGEN system as a "research infrastructure for systematic analysis" of multi-turn RL agents.
> > > The authors do cite RAGEN, but they dismiss it in a single line, which isn't even fully accurate, as the RAGEN paper discusses the need for fine-grained rewards. The authors fail to engage with this prior work and do not clearly differentiate their "three-pillar" framework conceptually from the RAGEN/StarPO framework.”*
> > >
> > > ***Response 3***: We appreciate the opportunity to clarify the positioning of our work relative to RAGEN (Wang et al., 2025). We acknowledge RAGEN as a valuable contribution to the engineering infrastructure of Agentic RL. However, we respectfully clarify that our “Three pillar” framework and RAGEN/StarPO framework have different scopes and address different research questions.
> > >
> > > To clarify why our work is distinct and necessary, we refer to the following table for a summary:
> > >
> > > | Feature | RAGEN (Wang et al., 2025\) | Our work |
> > > | :---- | :---- | :---- |
> > > | **Primary Contribution** | **Infrastructure & Implementation:** A modular codebase (built on verl) and a specific algorithm variant (StarPO). | **Design Methodology:** Provides recipes for Environment Complexity, RL policy, and Reward granularity. |
> > > | **Environment Scope** | **MDPs:** Focuses on standard RL tasks (Sokoban, Frozen Lake, Bandits) | **Semantic POMDPs:** Explicitly targets partial observability in high-fidelity language understanding tasks (TextWorld, ALFWorld, SWE-Gym). |
> > > | **Algorithmic Scope** | **Single Method (StarPO):** Formulates multi-turn RL as a specific trajectory-level PPO with thought tokens. | **Incorporating multiple RL algorithms:** Evaluates standard algorithms (PPO, GRPO, Reinforce++, RLOO) across different agentic tasks and rewards granularity. |
> > > | **Reward Scope** | **Discussing the need of dense reward:** Not provides entry points in their framework to integrate dense reward | **Implementing and analyzing dense reward:** Provides implementation of verified vs. model-based, sparse vs. dense reward, as well as their comparisons. |
> > >
> > > **First, the framework scope and contribution are different.** RAGEN (Wang et al., 2025) provides a software stack (similar to existing library as **verl**) and a specific algorithm formulation (StarPO). In contrast, our paper is **infrastructure-agonistic**; it means whether a practitioner uses RAGEN, verl, custom PyTorch code, or our codebase, they still face the **design questions** we answer: *How do I define environment complexity? What SFT:RL ratio to select under a fixed budget? What reward granularity should I use? Etc.* Our framework systematically analyzes these design choices and provides some guidance to practitioners.
> > >
> > > **Second, we study multi-turn RL on broad algorithms, beyond StarPO.** The reviewer cites StarPO as a framework; we view StarPO as **a specific multi-turn formulation of PPO that incorporates thinking steps**. There are two things we want to emphasize:
> > > 1. Regarding the novelty of reasoning of RAGEN (Wang et al., 2025): We note that "thinking/reasoning" tokens are now standard in modern RL (e.g., DeepSeek-R1). Our framework naturally treats reasoning as part of the trajectory not counting towards action tokens, within the formulation of a standard POMDP. We do not require specialized “StarPO” formulation on our agentic tasks.
> > > 2. Regarding our analysis of multiple RL algorithms: While RAGEN (Wang et al., 2025) promotes StarPO, our work systematically evaluates multiple RL algorithms (PPO, RLOO, GRPO, Reinforce++) across different task domains, and across different reward granularity.
> > >
> > > **Third, we formulate the agentic space as POMDP**. RAGEN (Wang et al., 2025) uses **stylized MDP**, where the state is fully observable. Our formulation targets general POMDP in semantic space (TextWorld, ALFWorld, SWE-Gym) where the true state is hidden. This necessitates our “Environment Pillar” metrics including spatial and object complexity, which is not required in RAGEN’s grid-worlds but crucial for our semantic language understanding tasks.
> > >
> > > **Lastly**, even though RAGEN mentioned the need for dense reward, they do not provide implementation entry points nor analysis on dense rewards. **Our work implements and analyzes the impacts of dense rewards on RL training**. More specifically, we provide both implementation and analysis on sparse vs. dense verified rewards, as well as verified vs. model-base rewards.
> > >
> > > We will revise our text in the final version to clarify the distinction between RAGEN and our work by positioning RAGEN as a combination of infrastructure + StarPO framework, while positioning our work as the methodological guide for designing the agents that run on any such infrastructure.

---

> ### Author Response · Authors · 2025-12-03
> **Response to Reviewer WXhe - 5**
>
> ***Major Weakness 4***: *“No SOTA Baseline Comparisons. For a paper that claims to be a "guide," it does a poor job of benchmarking its own "recipe." In Table 5 (ALFWorld) and Table 6 (SWE-Gym), the results are presented in a vacuum.*
>
> *Where is the comparison to the SOTA on ALFWorld?
> SPA-RL (Choudhury et al., 2025) reported 79.1% success on unseen ALFWorld tasks. This paper's best result in Table 5 is 74% (SFT+PPO).
> GiGPO (Feng et al., 2025) also reported a >12% gain over GRPO on ALFWorld.
> The paper presents a recipe that is sub-SOTA and doesn't even acknowledge, let alone discuss, why. This is a critical omission.”*
>
> ***Response 4***: We thank the reviewer for raising the comparison with SOTA models. However, there is **a critical difference in the task space** that makes a direct comparison with the reported numbers of SPA-RL and GiGPO misleading.
>
> As mentioned in our paper text (line 166-169): *“Critically, unlike traditional RL settings
> that provide both observations and admissible action lists (reducing the problem to action selection) our agents must generate executable natural language commands from environment observations alone, without action hints…”*, we clarify that there are no “**admissible actions**” provided at each step in the observation in our tasks. In contrast, **SPA-RL and GiGPO use admissible actions** (a list of all valid commands provided to the agent at every step).
>
> Here is a toy example showcasing the difference:
>
> **Without admissible actions (Ours)**:
>
> *State n: You are in the bathroom. You face the tabletop 5. You see sink 1, toothbrush 2, towel 3.*
>
> *Action n: take toothbrush 2 from tabletop 5*
>
> **With admissible actions (SPA-RL and GiGPO)**:
>
> *State n: You are in the bathroom. You face the tabletop 5. You see sink 1, toothbrush 2, towel 3. Your admissible actions are: take toothbrush 2 from tabletop 5, take towel 3 from tabletop 5, go to toilet 1, go to shower 4, ……*
>
> *Action n: take toothbrush 2 from tabletop 5*
>
> The problem is reduced to action selection (easier) if allowing admissible actions. Our paper focuses on open-ended language generation in agentic tasks (harder), so comparing our score (74%) against their score (79%) is not an apples-to-apples comparison.
>
> To further address the reviewer’s concern and provide a relatively fairer comparison, we re-ran our best recipe (SFT+PPO) on ALFWorld tasks with admissible actions enabled. The results are presented in the following table. ***Note that due to computation and time limitations, we ran our new experiments on 16 exploration steps, and RL is not fully converged when we report our answers. That being said, the performance may still improve over time.***
>
> First, our best result (84%) so far beats SPA-RL (79%). Under significantly smaller exploration step sizes (16 vs. 50), and without the fancy prompts (the baseline of GiGPO is 5% higher than our baseline), our best result (84%) so far still achieves very competitive performance as the GiGPO’s (91%).
>
> **In summary, our recipe is not subSOTA. The perceived gap was due to the difference in problem difficulty.**
>
> ***Table WXhe.4: Performance of Qwen-2.5-7B model trained using different methods on ALFWorld unseen tasks. * means reporting numbers from our paper. # means reporting numbers from other papers.***
> | Method | Protocol | Base model | Trained model | Exploration step |
> | :---- | :---- | :---- | :---- | :---- |
> | SPA-RL  | Action selection | Not reported | 0.791\# (best) | Not reported |
> | GiGPO  | Action selection | 0.148\# | 0.908\# (best) | 50 |
> | Ours (SFT only)  | Action selection | 0.107 | 0.55 | 16 |
> | Ours (SFT \+ RL)  | Action selection | 0.107 | 0.84 | 16 |
> | Ours (SFT only)  | Generation | 0.0\* | 0.47\* | 16 |
> | Ours (SFT \+ RL)  | Generation | 0.0\* | 0.74\* / 0.8\* (best) | 16 |

---

> > ### Author Response · Authors · 2025-12-03
> > **Response to Reviewer WXhe - 6**
> >
> > ***Question 1 (Reward Pillar)***: *“Your analysis in Section 7 is limited to sparse vs. built-in dense rewards. Can you please justify why you completely omitted any discussion or comparison with the entire line of work on learned or decomposed step-wise rewards, such as StepAgent (Deng et al., 2024), GiGPO (Feng et al., 2025), and SPA-RL (Choudhury et al., 2025)? How can this be a "practitioner's guide" if it ignores the SOTA reward-design techniques?”*
> >
> > ***Answer 1***: As detailed in [***Response 1***](https://openreview.net/forum?id=K6T0o875zF&noteId=DCFAJme0pN), we prioritize reproducible methods over inaccessible ones (StepAgent and SPA-RL have no code/checkpoints). We have now added experiments in ***new Section 7.2*** in new PDF to prove that our findings on reward granularity hold in complex tasks, using methods practitioners can actually implement.
> >
> > ----------------------------------------------------
> >
> > ***Question 2 (Policy Pillar - Algos)***: *“This is a direct question about novelty. Given that arXiv:2507.14897v1 already provides an empirical comparison of PPO, GRPO, and RLOO, what new insight does your Section 6.2 analysis provide that is not already present in that work? To be very clear, you need to explain what your contribution is over and above that paper.”*
> >
> > ***Answer 2***: As detailed in [***Response 2***](https://openreview.net/forum?id=K6T0o875zF&noteId=lh1Oj7ITdH), our work investigates distinct research questions tailored to long-horizon agents. The cited paper focuses on ​​Code Interpreter Math short-horizon tasks. In fact, we show that conclusions drawn from short-horizon Math tasks (where RLOO works) do not transfer to long-horizon Agentic POMDPs.
> >
> > -------------------------------------------------
> >
> > ***Question 3 (Policy Pillar - Ratios)***: *“Similarly, how do you position your findings on the SFT:RL ratio (Sec 6.1) against the large-scale, 1,370-run study in arXiv:2507.04103v1? Are your findings (from ~6 runs) simply a small-scale confirmation, or do you arrive at a different conclusion?”*
> >
> > ***Answer 3***: As detailed in [***Response 2***](https://openreview.net/forum?id=K6T0o875zF&noteId=lh1Oj7ITdH), the cited paper focuses on compute scaling. Our contribution is on data cost constraints (modeling the 10x cost difference between SFT and RL) and easy-to-hard environment complexity generalization.
> >
> > -------------------------------------------------
> >
> > ***Question 4 (Framework Claim)***: *“Can you please elaborate in detail on the conceptual novelty of your "three-pillar" framework compared to the "systematic analysis" framework proposed by RAGEN (Wang et al., 2025)? The current dismissal in your related work section is insufficient.”*
> >
> > ***Answer 4***: As detailed in [***Response 3***](https://openreview.net/forum?id=K6T0o875zF&noteId=8ZpxNrFDHa), we show the distinction between RAGEN and our work by positioning RAGEN as a combination of infrastructure + StarPO framework, while positioning our work as the methodological guide for designing the agents that run on any such infrastructure.

---

### Official Review · Reviewer_vYsC · 2025-11-05

**Soundness:** 3
**Presentation:** 4
**Contribution:** 2
**Rating:** 6
**Confidence:** 4

**Summary:**

The paper provides an empirical recipe for training LLM agents using multi-turn RL. Specifically, the paper considers the impact of different designs of environment, reward, and training algorithm on overall performance, ultimately showing that dense turn-level rewareds and prior SFT training greatly accelerate RL training and reduce sample complexity.

**Strengths:**

1. The paper does a systematic evaluation of multiple facets of RL training to create a comprehensive blueprint that will be generally useful to practitioners trying to train LLM agents.

2. The evaluation is done over three very different textual domains, which greatly increases the generalizability of the proposed blueprint.

3. The analysis on the usefulness of prior SFT training and the optimal SFT:RL data ratio will be useful for other applications.

**Weaknesses:**

1. I found the analysis of biased (PPO, GRPO) vs unbiased (RLOO) algorithms to be insufficient. The authors claim the improvement is due to multi-turn training, but it seems evident that algorithm choice has a noticeable effect. However, this difference in performance between different algorithms is not analyzed further or explained.

2. The finding that dense reward is better than sparse rewards was only based on evidence from simple tasks. I believe testing dense vs. sparse rewards on the complex tasks is important to truly determine the usefulness of converting sparse to dense rewards.

3. The authors claim that SFT training can cause rapid collapse when fine-tuned on a different environment. However, the evidence for this is on one particular policy size and architecture. It is unclear if increased model capacity still results in the same instability.

**Questions:**

1. What is the effect of prior SFT training when the data is not only expert demonstrations, but instead consists of some noisy or suboptimal examples?

2. The paper currently shows that pretraining on one particular complex task generalizes poorly to other complex tasks. Does the specific task used have effect on the results? I am curious of what would happen if the pretraining was only on specific skills that ultimately needed to be composed.

3. The authors showed that PPO outperforms RLOO but does not give a reason why. It could be in the scaling of the rewards, as RLOO is more sensitive to the magnitude of rewards. Could perhaps increasing the scaling of the reward make the performance of RLOO match that of PPO?

---

> ### Author Response · Authors · 2025-12-03
> **Response to Reviewer vYsC - 1**
>
> ***Comment 1***: *“I found the analysis of biased (PPO, GRPO) vs unbiased (RLOO) algorithms to be insufficient. The authors claim the improvement is due to multi-turn training, but it seems evident that algorithm choice has a noticeable effect. However, this difference in performance between different algorithms is not analyzed further or explained.”*
>
> ***Response 1***: We agree that the mechanism behind the PPO, GRPO vs. RLOO performance gap needs further analysis. We have added the **training curves of PPO vs. RLOO** in different TextWorld tasks, **across 1.5B and 7B models**, presented in ***new Figure 6, 7 in Appendix***.
>
> As shown in both Figures, PPO exhibits a stable and consistent increase in reward, while RLOO exhibits higher variance and slower convergence. PPO has higher final converging reward than RLOO for both 1.5B and 7B models. We further explain this as the **variance-bias tradeoff** in multi-turn scenarios:
> - RLOO is an unbiased method, which relies on the mean of N samples as a baseline. In long-horizon tasks, the variance of the entire trajectory return is naturally high because it accumulates from each step. So the baseline becomes a relatively noisy estimator for a specific action at step t, making credit assignment noisy over the multi-turn context.
> - PPO is a biased heuristics-based method, which uses a value function as the baseline. Although it introduces bias, it significantly reduces variance. The value function provides a step-specific baseline V(s_t) (for each token) that estimates a more fine-grained future return from the specific state. So with the localized baseline, PPO allows for more stable credit assignment.
>
> The final performance of PPO (88%) and RLOO (51%) in TextWorld tasks also shows that the variance reduction from the critic proves more critical for learning in multi-turn tasks than the unbiased nature of RLOO.

---

> > ### Author Response · Authors · 2025-12-03
> > **Response to Reviewer vYsC - 2**
> >
> > ***Comment 2***: *“The finding that dense reward is better than sparse rewards was only based on evidence from simple tasks. I believe testing dense vs. sparse rewards on the complex tasks is important to truly determine the usefulness of converting sparse to dense rewards.”*
> >
> > ***Response 2***: We completely agree with the need to test dense vs. sparse reward on more complex tasks. To address this, we conduct additional experiments in **SWE-Gym**, which represents a more complex task space (coding, tool-use, long horizon, closer to realworld tasks) than TextWorld and ALFWorld.
> >
> > We further compare the following reward types in SWE-Gym:
> > 1. **Binary verified reward (sparse)**: Agent receives +1 at the end only if it passes the test suite.
> > 2. **Ratio-based verified reward (dense)**: Agent receives reward proportional to the percentage of unit tests passed so far.
> >
> > As shown in the following table (same as ***new Table 12*** in new PDF), the ratio-based reward achieves 22% success rate, compared with the binary reward at 4.2% in SWE-Gym tasks. This is consistent with our findings in Section 7.1 that carefully designed dense, process-oriented rewards are essential for training agents via RL in complex environments.
> >
> > ***Table vYsC.1: Performance of Qwen3-8B model trained using multi-turn GRPO on SWE-Gym tasks with different reward types, including verified (binary, ratio-based) and model-based (CodeRM-8B as judge, GPT-4.1 as judge). * means reporting numbers from our paper.***
> > | Tasks / rewards | Base model\* | Verified (binary) | Verified (ratio) | Model judge (CodeRM-8B) | Model judge (GPT-4.1) |
> > | :---- | :---- | :---- | :---- | :---- | :---- |
> > | SWE-Gym (GRPO) | 0.04\* | 0.042 (+0.002) | 0.22 (+0.18) | 0.072 \+ (0.032) | 0.093 \+ (0.053) |

---

> > > ### Author Response · Authors · 2025-12-03
> > > **Response to Reviewer vYsC - 3**
> > >
> > > ***Comment 3***: *“The authors claim that SFT training can cause rapid collapse when fine-tuned on a different environment. However, the evidence for this is on one particular policy size and architecture. It is unclear if increased model capacity still results in the same instability.”*
> > >
> > > ***Response 3***: To verify if this rapid collapse is specific to model size, we add additional experiments on fine tuning on one environment and train RL on another environment, with different model sizes: 1.5B and 7B.
> > >
> > > The following table shows that **even with different model sizes, both models exhibit similar trend**: SFT on one domain fail to guarantee multi-turn RL performance in another domain, even tho the two domains have some similarities (TextWorld focuses on completing quests in a map while ALFWorld focuses on finishing household tasks in one room). The common failure across different model sizes indicates that **the SFT collapse in agentic environments is due to prior policy mismatch, not a lack of model capability.**
> > >
> > >
> > > ***Table vYsC.2: Results of models with different sizes (Qwen-2.5-1.5B and 7B), tested on the original domain after SFT, and based on that SFT model, further do RL on a new domain, tested on the new domain.***
> > > | SFTed Models | Result on original domain after SFT | Results on new domain after RL |
> > > | :---- | :---- | :---- |
> > > | Qwen-2.5-**1.5B** SFTed on ALFWorld | 0.16 (on ALFWorld) | 0.0 (on TextWorld) |
> > > | Qwen-2.5-**7B** SFTed on ALFWorld | 0.47 (on ALFWorld) | 0.06 (on TextWorld) |
> > > | Qwen-2.5-**1.5B** SFTed on TextWorld | 0.62 (on TextWorld) | 0.0 (on ALFWorld) |
> > > | Qwen-2.5-**7B** SFTed on TextWorld | 0.79 (on TextWorld) | 0.0 (on ALFWorld) |

---

> > > > ### Author Response · Authors · 2025-12-03
> > > > **Response to Reviewer vYsC - 4**
> > > >
> > > > ***Question 1***: *“What is the effect of prior SFT training when the data is not only expert demonstrations, but instead consists of some noisy or suboptimal examples?”*
> > > >
> > > > ***Answer 1***: This is an interesting question. In this paper, we focus on expert demonstrations because obtaining high-quality “suboptimal” data is surprisingly difficult. Collecting human trajectories is costly and generating them synthetically is **non-trivial** because current SOTA models like GPT-4.1 already solves TextWorld and most SWE-Gym tasks.
> > > >
> > > > We agree that analyzing the impact of noisy SFT data is an interesting research direction. Specifically, comparing expert trajectories derived from heuristic search versus those derived from human mental models (which may include backtracking or suboptimal yet insightful steps) could reveal important insights into the SFT + RL training paradigm.
> > > >
> > > > --------------------------------
> > > >
> > > > ***Question 2***: *“The paper currently shows that pretraining on one particular complex task generalizes poorly to other complex tasks. Does the specific task used have effect on the results? I am curious of what would happen if the pretraining was only on specific skills that ultimately needed to be composed.”*
> > > >
> > > > ***Answer 2***: This is an insightful question regarding the impact of “pre-training on specific skills” rather than full tasks on improving the robustness of RL training. We agree that decomposing a specific task into atomic skills and training SFT specifically on that may improve robustness. However, we want to emphasize that effective skill decomposition is non-trivial. Determining exactly which atomic skills are necessary for a complex task, and how much pre-training each requires, is an open research question. We think this skill-based SFT -> RL pipeline is a promising next step for the field.
> > > >
> > > > --------------------------------------------
> > > > ***Question 3***: *“The authors showed that PPO outperforms RLOO but does not give a reason why. It could be in the scaling of the rewards, as RLOO is more sensitive to the magnitude of rewards. Could perhaps increasing the scaling of the reward make the performance of RLOO match that of PPO?”*
> > > >
> > > > ***Answer 3***: We clarify that in all our experiments, **rewards are normalized to [0, 1] range for all algorithms** to ensure fair comparison. Therefore, the performance gap is likely not due to the scalar magnitude differences.
> > > >
> > > > However, we acknowledge the reviewer’s intuition: since RLOO is an unbiased estimator with higher variance, there likely exists a specific set of hyperparameters (e.g. significantly larger sample size, specific learning rates, or reward scaling factors) that could stabilize RLOO to match PPO’s performance. Our results highlight that PPO provides greater stability and the performance out of the box with default parameters, while RLOO requires more sensitive     parameter tuning to handle the variance of multi-turn agentic tasks.

---

### Official Review · Reviewer_N61V · 2025-11-05

**Soundness:** 2
**Presentation:** 2
**Contribution:** 2
**Rating:** 4
**Confidence:** 3

**Summary:**

This paper provides a systematic study of design choices for multi-turn agentic reinforcement learning. The authors first distinguish their setting from pseudo multi-turn methods and decompose their design space into three key components: environment, policy, and reward. For the environment, they analyze the impact of environment complexity and how agents trained via multi-turn RL generalize to harder complexities. For the policy, they discuss the influence of initial policies, the impact of different SFT/RL data ratios, and the effectiveness of different RL algorithms in multi-turn settings. For the reward, they study the impact of sparse/dense reward. Through extensive experiments on TextWorld, ALFWorld, and SWE-Gym, the authors derive a practical training recipe for multi-turn agentic RL with respect to the above three components.

**Strengths:**

1. This paper is well written and easy to follow, with a good overall structure and clear motivation.

2. Figure 1 provides an easy-to-understand overview of the work, making it easy for readers to understand the overall idea.

3. The paper clearly distinguishes different types of “multi-turn” RL, making the scope of the work precise and well defined.

4. This paper conducts a large number of experiments to validate many important design choices for agentic RL, which could be very helpful for the research community.

**Weaknesses:**

1. Some conclusions seem to be closely tied to the experimental environment, therefore limited in their generalizability. For example, the concept of spatial complexity seems to be closely related to embodied reasoning, but this concept may not be always available in other agentic environments.

2. The authors try to cover many different design choices in their paper. However, some of them are supported by only one experiment in a single environment, making it hard to draw general conclusions from these experiments. For example, the experiment on sparse/dense reward is only conducted in TextWorld.

3. This paper is well written in terms of text, but other aspects of the presentation could be improved. For example, Tables 5 and 6 could be merged into one table to make the presentation more concise, as they discuss the same topic. Additionally, the corresponding table for lines 374-377 seems to be missing. It would also be better to increase the diversity of the presentation, all experimental results are now presented as tables. Some of them could be converted into figures to make the paper more visually appealing.

4. In general, I appreciate the effort the authors have made to cover various design choices and supporting their claims with a substantial number of experiments. However, as a PRACTITIONER’S GUIDE (as indicated by the title of this paper), I believe the number of experiments still needs to be increased both in terms of breadth and depth. Specifically, more combinations of factors, such as different RL algorithms, experimental environments, model families, and model sizes, should be explored to provide more solid support for their claims.

**Questions:**

1. The performance of Qwen-1.5B in w2-o3-q4 is different in Tables 1 and 2 (0.17 and 0.15). Is that just a typo, or do I misunderstand the setting?

2. What's the base model for Tables 4, 5, and 6?

---

> ### Author Response · Authors · 2025-12-03
> **Response to Reviewer N61V - 1**
>
> We thank the reviewer for their feedback and for acknowledging the extensive efforts behind our experiments.
>
> Below, we address your comments point-by-point:
>
> ----------------------------------
> ***Comment 1***: *“Some conclusions seem to be closely tied to the experimental environment, therefore limited in their generalizability. For example, the concept of spatial complexity seems to be closely related to embodied reasoning, but this concept may not be always available in other agentic environments.”*
>
> ***Response 1***: We agree that spatial complexity is domain-specific for embodied tasks. To demonstrate that our findings on **environment complexity** hold broadly, we introduce equivalent complexity metrics for **coding environments** as well. We test on **SWE-Gym** tasks, a domain distinct from embodied or text games.
>
> As shown in the following tables (same as ***new Table 2*** in new PDF), we categorized SWE-Gym tasks into Easy, Medium, and Hard based on two new complexity metrics:
>
> 1. **Solution Complexity**: Defined as the number of lines in the ground truth solution patch to a code problem, indicating the **implementation effort** of a code agent to write the solution.
> 2. **Test Complexity**: Defined as the size of the test suite. More specifically, it is the number of all “Fail to pass” and “Pass to pass” unit tests per test suite, indicating the **verification rigorousness** of a coding task.
>
> Our metrics are validated in the tables where the base model’s performance decreases as environment complexity increases. And after RL training, all agents show improvement over the base performance.
>
> ***Table N61V.1: Task completion ratio comparison between the base Qwen3-8B model and the model trained using Multi-turn GRPO, across three train datasets of different environment complexities constructed in terms of Metric A: the solution size/lines complexity. The train datasets are of the same size.***
> | Metric A: tasks w/ varying lines of patch | Qwen3-8B (base) | Qwen3-8B (GRPO) |
> | :---- | :---- | :---- |
> | easy (≤ 8 lines)  | 0.048 | 0.072 |
> | medium (9 \- 31 lines) | 0.048 | 0.109 |
> | hard (\> 31 lines) | 0.018 | 0.096 |
>
>
> ***Table N61V.2: Task completion ratio comparison between the base Qwen3-8B model and the model trained using Multi-turn GRPO, across three train datasets of different environment complexities constructed in terms of Metric B: test suite size/tests complexity. The train datasets are of the same size.***
> | Metric B: tasks w/ varying test suite sizes | Qwen3-8B (base) | Qwen3-8B (GRPO) |
> | :---- | :---- | :---- |
> | easy (≤ 17 tests) | 0.04 | 0.133 |
> | medium (18 \- 53 tests) | 0.028 | 0.075 |
> | hard (\> 53 tests) | 0.017 | 0.105 |
>
>
> In addition, we provide **cross-complexity generalization** results in the following tables (as well as ***new Table 6*** in new PDF). The results show that code agents trained in simpler environments achieve improvement in harder environments. For example, the code agent trained on Easy tasks (tests complexity metric) gained 4.8% and 3.6% in Medium and Hard tasks, respectively. It mirrors our findings in paper section 5.2 where we show that TextWorld agents trained with multi-turn RL have generalization abilities that transfer from simple to complex environments.
>
> ***Table N61V.3: Performance of the Qwen3-8B model trained using Multi-turn GRPO, trained across three train datasets of different environment complexities and tested on all environment complexities. In terms of Metric A: the solution size/lines complexity.***
> | Trained on/ Tested on  |easy | medium | hard|
> | :---- | :---- | :---- | :---- |
> | easy | 0.072 (+0.024) | 0.048 (+0.0) | 0.028 (+0.01) |
> | medium | 0.161 (+0.113) | 0.109 (+0.061) | 0.055 (+0.037) |
> | hard | 0.082 (+0.034) | 0.091 (+0.043) | 0.096 (+0.078) |
>
> ***Table N61V.4: Performance of the Qwen3-8B model trained using Multi-turn GRPO, trained across three train datasets of different environment complexities and tested on all environment complexities. In terms of Metric B: test suite size/tests complexity.***
> | Trained on/ Tested on  | easy | medium| hard |
> | :---- | :---- | :---- | :---- |
> | easy | 0.133 (+0.093) | 0.076 (+0.048) | 0.053 (+0.036) |
> |medium | 0.082 (+0.042) | 0.075 (+0.047) | 0.042 (+0.025) |
> | hard | 0.1 (+0.06) | 0.062 (+0.034) | 0.105 (+0.088) |
>
> **In summary, the new results confirm that, with carefully designed environment complexity metrics, our paper’s insights transfer to non-embodied domains such as coding.**

---

> > ### Author Response · Authors · 2025-12-03
> > **Response to Reviewer N61V - 2**
> >
> > ***Comment 2***: *“The authors try to cover many different design choices in their paper. However, some of them are supported by only one experiment in a single environment, making it hard to draw general conclusions from these experiments. For example, the experiment on sparse/dense reward is only conducted in TextWorld.”*
> >
> > ***Response 2***: We have significantly expanded our experiments in breadth and depth and validated our claims across new environments, algorithms, and rewards. You can find the results in the new PDF (marked blue) with references in the general response.
> >
> > To specifically address your concerns for the experiment on sparse/dense reward, we extend our analysis to **SWE-Gym**. In TextWorld experiments, we observe that heuristics-based dense rewards outperformed sparse rewards. We extend this setup in the coding domain by comparing:
> > 1. **Binary verified reward (sparse)**: Agent receives +1 at the end only if it passes the test suite.
> > 2. **Ratio-based verified reward (dense)**: Agent receives reward proportional to the percentage of unit tests passed so far.
> >
> > As shown in the following table (same as **new Table 12** in new PDF), the ratio-based reward achieves 22% success rate, compared with the binary reward at 4.2%. This is consistent with our findings in Section 7.1 that carefully designed dense, process-oriented rewards are essential for training agents via RL in complex environments.
> >
> > ***Table N61V.5: Performance of Qwen3-8B model trained using multi-turn GRPO on SWE-Gym tasks with different reward types, including verified (binary, ratio-based) and model-based (CodeRM-8B as judge, GPT-4.1 as judge). * means reporting numbers from our paper.***
> > | Tasks / rewards | Base model\* | Verified (binary) | Verified (ratio) | Model judge (CodeRM-8B) | Model judge (GPT-4.1) |
> > | :---- | :---- | :---- | :---- | :---- | :---- |
> > | SWE-Gym (GRPO) | 0.04\* | 0.042 (+0.002) | 0.22 (+0.18) | 0.072 \+ (0.032) | 0.093 \+ (0.053) |

---

> ### Author Response · Authors · 2025-12-03
> **Response to Reviewer N61V - 3**
>
> ***Comment 3***: *“This paper is well written in terms of text, but other aspects of the presentation could be improved. For example, Tables 5 and 6 could be merged into one table to make the presentation more concise, as they discuss the same topic. Additionally, the corresponding table for lines 374-377 seems to be missing. It would also be better to increase the diversity of the presentation, all experimental results are now presented as tables. Some of them could be converted into figures to make the paper more visually appealing.”*
>
> ***Response 3***: We thank the reviewer for these suggestions to improve readability.
> - Tables: We will merge Tables 5 and 6 in the final version
> - Missing data: We didn’t see cross-domain generalization of models, and the success rate is zero due to policy collapse. That’s - why we explain in text “cross-domain priors cause rapid policy collapse during multi-turn RL” as the description of the table.
> - Figures: We will convert the SFT vs. RL efficiency analysis to a trade-off curve figure (cost vs. success rate) for the final version.
>
> ---------------------------------------------------------------
> ***Comment 4***: *“In general, I appreciate the effort the authors have made to cover various design choices and supporting their claims with a substantial number of experiments. However, as a PRACTITIONER’S GUIDE (as indicated by the title of this paper), I believe the number of experiments still needs to be increased both in terms of breadth and depth. Specifically, more combinations of factors, such as different RL algorithms, experimental environments, model families, and model sizes, should be explored to provide more solid support for their claims.”*
>
> ***Response 4***: We have significantly expanded the experiments in breadth and depth, all included in the general rebuttal response. Here is a summary of our revisions (same as what I put in general response):
> 1. **Expanded environment analysis**: We add SWE-Gym experiments to validate our findings of environment complexity (***new Table 2***) and the impact of agent’s exploration steps (max steps to take per task) during training (***new Table 4***). We further analyze the easy-to-hard generalization performance for models trained on different environment complexities in SWE-Gym tasks (***new Table 6***)
>
> 2. **Expanded policy analysis**: We implement and evaluate additional RL algorithms (multi-turn version), including Reinforce++, and GRPO to provide a comprehensive analysis of the impacts of RL algorithms on multi-turn training (***new Table 10***). We also provide training reward curves for comparison of PPO and RLOO algorithms on TextWorld tasks (***new Figure 6, 7 in Appendix***).
>
> 3. **Expanded reward shaping analysis**: We add SWE-Gym experiments to reward analysis and provide an extra dimension of reward comparison. More specifically, we compare binary vs. ratio-based reward as well as ratio-based verified reward vs. model-based reward (CodeRM-8B, GPT-4.1) in the coding domain (***new Table 12***)
>
> ---------------------------------------------------
> ***Question 1***: *“The performance of Qwen-1.5B in w2-o3-q4 is different in Tables 1 and 2 (0.17 and 0.15). Is that just a typo, or do I misunderstand the setting?”*
>
> ***Answer 1***: Thank you for pointing this out. It is **not a typo** because the Qwen-2.5-1.5B model in Tables 1 and 2 are tested under **different exploration sizes (max steps per episode)**. Therefore, the result from Table 2 is lower than Table 1 since its exploration size is smaller (12 vs. 16).
>
> ---------------------------------------------------
>
> ***Question 2***: *“What's the base model for Tables 4, 5, and 6?”*
>
> ***Answer 2***: The base model is Qwen2.5-1.5B-Instruct.

---

### Author Response · Authors · 2025-12-03
**General response to all reviewers**

We thank the reviewers for their constructive feedback and for recognizing the timeliness and structural clarity of our paper. We agree that a robust guide requires rigorous empirical backing. In response to the reviewers’ suggestions (particularly regarding experimental breadth and depth), we have expanded the scope of our paper with **three major sets of new experiments**:

1. **Expanded environment analysis**: We add SWE-Gym experiments to validate our findings of environment complexity (***new Table 2***) and the impact of agent’s exploration steps (max steps to take per task) during training (***new Table 4***). We further analyze the easy-to-hard generalization performance for models trained on different environment complexities in SWE-Gym tasks (***new Table 6***)

2. **Expanded policy analysis**: We implement and evaluate additional RL algorithms (multi-turn version), including Reinforce++, and GRPO to provide a comprehensive analysis of the impacts of RL algorithms on multi-turn training (***new Table 10***). We also provide training reward curves for comparison of PPO and RLOO algorithms on TextWorld tasks (***new Figure 6, 7 in Appendix***).

3. **Expanded reward shaping analysis**: We add SWE-Gym experiments to reward analysis and provide an extra dimension of reward comparison. More specifically, we compare binary vs. ratio-based reward as well as ratio-based verified reward vs. model-based reward (CodeRM-8B, GPT-4.1) in the coding domain (***new Table 12***)

All modifications on the original PDF are **marked blue** in the newest PDF. We also provide tables in the responses for convenience.

---

### Meta-Review · Area_Chair_HwS1 · 2026-01-02

**Summary:**

This work conducts an empirical investigation of multiturn agentic reinforcement learning. It is organized along three axes: environment complexity, reward, and algorithm. It studies these different axes and concludes with a practical recipe. Overall, the reviews tended towards reject (4446). Some reviewers cited insufficient empirical depth, and asked for more comparisons along the different axes (more algorithms, or environments). In the rebuttal, the authors added some additional experiments to address these, for example, a new environment (SWE gym) and two extra RL algorithms. This may have caused some modest increases in the reviewer scores. However, I don't think any of the reviewers would have strongly advocated for the paper’s acceptance. Overall, I agree with the reviewers (especially iMhR) that the paper for the most part confirms phenomena that are already known. For example, the final conclusion is that: dense rewards perform better than sparse rewards; curricula that go from easy tasks to more complex tasks facilitate learning; and bootstrapped reinforcement learning algorithms are beneficial for longer-horizon tasks. These are well established facts in the RL literature, and are not surprising. The open question rather, is in **how** to define dense reward heuristics or learning curricula in an automatic way (without excessive human engineering or prior knowledge), rather than whether they are useful or not. The paper does not provide new insights into these more important questions, therefore I am recommending reject.

**Reviewer Concerns:**

The reviewer concerns around lack of empirical depth were partially addressed by the rebuttal. Namely, the authors added experiments with a new environment, and two new RL algorithms. However, all experiments were still with textual inputs only (one of the reviewers asked about multimodal inputs, which I agree would have strengthened the paper). I don't think the broader concerns about whether this paper provides new insights useful to the community have been addressed.

**Reviewer Scores:**

N61V: 4 -> 6

vYsC: 6 -> 6

WXhe: 4 -> 4

iMhR: 4 -> 4

---

### Decision · Program_Chairs · 2026-01-26

Reject